# FINDING ADVERSARIALLY ROBUST GRAPH LOTTERY TICKETS

## ABSTRACT

Graph Lottery Tickets (GLTs), comprising a sparse graph neural network (GNN) and a sparse input graph adjacency matrix, can significantly reduce the inference latency and compute footprint compared to their dense counterparts. However, their performance against adversarial structure perturbations remains to be fully explored. In this work, we first investigate the resilience of GLTs against different structure perturbation attacks and observe that they are vulnerable and show a large drop in classification accuracy. We then present an *adversarially robust graph sparsification (ARGS)* framework that prunes the adjacency matrix and the GNN weights by optimizing a novel loss function capturing the graph homophily property and information associated with both the true labels of the train nodes and the pseudo labels of the test nodes. By iteratively applying ARGS to prune both the perturbed graph adjacency matrix and the GNN model weights, we can find adversarially robust graph lottery tickets that are highly sparse yet achieve competitive performance under different untargeted training-time structure attacks. Evaluations conducted on various benchmarks, considering different poisoning structure attacks such as PGD, MetaAttack, the PR-BCD attack, and adaptive attacks, demonstrate that the GLTs generated by ARGS can significantly improve their robustness, even when subjected to high levels of sparsity.

## 1 INTRODUCTION

Graph neural networks (GNNs) (Hamilton et al., 2017; Kipf & Welling, 2016; Veličković et al., 2017; Zhou et al., 2020; Zhang et al., 2020) achieve state-of-the-art performance on various graph-based tasks like semi-supervised node classification (Kipf & Welling, 2016; Hamilton et al., 2017; Veličković et al., 2017), link prediction (Zhang & Chen, 2018), and graph classification (Ying et al., 2018). The success of GNNs is attributed to the neural message-passing scheme in which each node updates its feature by recursively aggregating and transforming the features of its neighbors. However, the effectiveness of GNNs when scaled up to large and densely connected graphs is impacted due to high training cost, inference latency, and substantial memory consumption. Unified graph sparsification (UGS) (Chen et al., 2021) addresses this concern by simultaneously pruning the input graph adjacency matrix and the GNN to show the existence of a graph lottery ticket (GLT), a pair of sparse graph adjacency matrix and GNN model, which can potentially accelerate inference without compromising model performance.

Recent studies reveal that GNNs are vulnerable to adversarial attacks (Dai et al., 2018; Wu et al., 2019; Zügner & Günnemann, 2019; Mujkanovic et al., 2022; Jin et al., 2020a). An adversarial attack introduces unnoticeable perturbations to the graph structure or node features. These perturbations increase the distribution shift between train nodes and test nodes, fooling the GNN to misclassify nodes in the graph (Li et al., 2023). Compared to node features, altering the graph structure has a more significant impact on the accuracy. To counter these attacks, many defense techniques have been developed. Some techniques improve the classification accuracy of GNNs by cleaning the perturbed graph structure (Wu et al., 2019; Entezari et al., 2020; Jin et al., 2020b; Deng et al., 2022; Zhu et al., 2021b); others improve the accuracy by modifying the GNN architecture (Zhang & Zitnik, 2020; Geisler et al., 2021; Zhu et al., 2019). Although GLTs demonstrate strong performance on original benign graph data, their performance in the presence of adversarial structure perturbations remains largely unexplored. Achieving adversarially robust GLTs (ARGLTs) can enable efficient GNN inference under adversarial threats.

To this end, we first empirically investigate the resilience of GLTs identified by UGS against different structure perturbation attacks (Zügner & Günnemann, 2019; Liu et al., 2019; Mujkanovic et al., 2022) showing that they are vulnerable. We then present ARGS (Adversarially Robust Graph Sparsification), an optimization framework that, given an adversarially perturbed graph, iteratively prunes the graph adjacency matrix and the GNN model weights to generate an adversarially robust graph lottery ticket (ARGLT) that achieves competitive classification accuracy while exhibiting high levels of sparsity. To the best of our knowledge, this is the first study on the adversarial robustness of GLTs. Adversarial attacks like the projected gradient descent (PGD) attack (Wu et al., 2019), the meta-learning-based graph attack (MetaAttack) (Zügner & Günnemann, 2019),

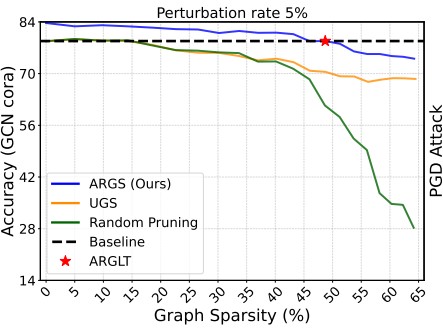

Figure 1: Comparison of different graph sparsification techniques in accuracy vs. graph sparsity. ARGS archives similar accuracy with 35% more sparsity for the Cora dataset under PGD attack.

and projected randomized block coordinate descent (PR-BCD) (Geisler et al., 2021), poison the graph structure by adding new edges or deleting existing edges, resulting in changes in the properties of the underlying graph. Often these attacks introduce most edge modifications around the training nodes (Li et al., 2023) while the local structure of the test nodes is less affected. Moreover, our analysis shows that, for homophilic graphs, adversarial edges are often introduced between nodes with dissimilar features. In contrast, for heterophilic graphs, adversarial edges are introduced between nodes with dissimilar neighborhood structures. We leverage this information to formulate new loss functions that better guide the pruning of the adversarial edges in the graph and the GNN weights. Additionally, we use self-learning to train the pruned GNNs on sparse graph structures, which improves the classification accuracy of the GLTs.

Our proposal is evaluated across various GNN architectures on both homophilic (Cora, citeseer, PubMed, OGBN-ArXiv) and heterophilic (Chameleon, Squirrel) graphs attacked by the PGD, MetaAttack, and PR-BCD attack (Geisler et al., 2021) for the node classification task. We also evaluate the proposed technique on adaptive attacks. By iteratively applying ARGS, ARGLTs can be broadly located across the 6 graph datasets with substantially reduced inference costs (up to 98% multiply-and-accumulate savings) and little to no accuracy drop. Figure 1 shows that for node classification on Cora attacked by the PGD attack, our ARGLT achieves similar accuracy to that of the full models and graphs even with high graph and model sparsity of 48.68% and 94.53% respectively. Compared to the GLT identified by UGS, our ARGLT achieves the same accuracy with 2.4× more graph sparsity and 2.3× more model sparsity.

## 2 RELATED WORK

**Graph Lottery Ticket Hypothesis.** The lottery ticket hypothesis (LTH) (Frankle & Carbin, 2018) conjectures that there exist small sub-networks, dubbed as lottery tickets (LTs), within a dense randomly initialized network, that can be trained in isolation to achieve comparable accuracy to their dense counterparts. UGS made it possible to extend the LTH to GNNs (Chen et al., 2021), showing the existence of GLTs that can accelerate GNN inference. A GNN sub-network along with a sparse graph is defined as a GLT if the sub-network with the original initialization trained on the sparsified graph has a matching test accuracy to the original unpruned GNN trained on the full graph. Specifically, during end-to-end training, UGS applies two differentiable binary mask tensors to the adjacency matrix and the GNN model weights, respectively. After training, the lowest-magnitude elements are removed and the corresponding mask location is updated to 0, eliminating the low-scored edges and weights from the adjacency matrix and the GNN, respectively. The sparse GNN weight parameters are then rewound to their original initialization. To identify the GLTs, the UGS algorithm is applied in an iterative fashion until pre-defined graph and weight sparsity levels are reached. Experimental results show that UGS can significantly trim down the inference computational cost without compromising predictive accuracy. In this work, we aim to find GLTs for datasets that have been adversarially perturbed. When we apply the UGS algorithm directly to the perturbed graphs, the performance accuracy of the GLTs is substantially low compared to their clean counterparts, calling for developing new optimization method to find adversarially robust GLTs.

**Adversarial Attacks on Graphs.** Adversarial attacks on graphs can be classified as poisoning attacks, perturbing the graph at train time, and evasion attacks, perturbing the graph at test time. Both poisoning and evasion attacks can be targeted or global attacks (Liu et al., 2019). A targeted attack deceives the model to misclassify a specific node (Zügner et al., 2018; Bojchevski & Günnemann, 2019). A global attack degrades the overall performance of the model (Zügner & Günnemann, 2019; Wu et al., 2019). Depending on the amount of information available, the existing attacks can be categorized into white-box attacks, practical black-box attacks, and restricted black-box attacks (Zügner et al., 2018; Chang et al., 2020). An attacker can modify the node features, the discrete graph structure, or both. Different attacks show that structure perturbation is more effective, when compared to modifying the node features. Examples of global poisoning attacks include the MetaAttack (Zügner & Günnemann, 2019), PGD attack (Wu et al., 2019), and PR-BCD attack (Geisler et al., 2021). Gradient-based attacks like PGD and MetaAttack treat the adjacency matrix as a parameter tensor and modify it via scaled gradient-based perturbations that aim to maximize the loss, thus resulting in degradation of the GNN prediction accuracy. PR-BCD (Geisler et al., 2021) is a more scalable first-order optimization attack that can scale up to large datasets like OGBN-ArXiv (Hu et al., 2020). Global poisoning attacks are highly effective in reducing the classification accuracy of different GNNs and are typically more challenging to counter, since they modify the graph structure before training (Zhu et al., 2021a). Therefore, we consider global graph structure poisoning attacks and ARGLTs under the poisoning setting.

**Defenses on Graphs.** Several approaches have been developed to combat adversarial attacks on graphs (Tang et al., 2020; Entezari et al., 2020; Zhu et al., 2019; Jin et al., 2020b; Zhang & Zitnik, 2020; Wu et al., 2019; Deng et al., 2022; Zhou et al., 2023). Many of these techniques try to improve the classification accuracy by preprocessing the graph structure, i.e., they detect the potential adversarial edges and assign lower weights to these edges, or even remove them. Jaccard-GCN (Wu et al., 2019) removes all the edges between nodes whose features exhibit a Jaccard similarity below a certain threshold. SVD-GCN (Entezari et al., 2020) replaces the adjacency matrix with a low-rank approximation, since many real-world graphs are low-rank and attacks tend to disproportionately affect the high-frequency spectrum of the adjacency matrix. ProGNN (Jin et al., 2020b) leverages low rank, sparsity, and feature smoothness properties of graphs to clean the perturbed adjacency matrix. GARNET (Deng et al., 2022) combines spectral graph embedding with probabilistic graphical models to recover the original graph topology from the perturbed graph. GNNGuard (Zhang & Zitnik, 2020) learns weights for the edges in each message passing aggregation via cosine-similarity and penalizes the adversarial edges either by filtering them out or by assigning less weight to them. Other techniques try to improve the GNN performance by enhancing model training through data augmentation (Li et al., 2022; Feng et al., 2020), adversarial training (Wu et al., 2019), self-learning (Li et al., 2023), robust aggregate functions (Geisler et al., 2021), or by developing novel GNN layers (Zhu et al., 2019). Differently from these works, GCN-LFR (Chang et al., 2021), a spectral-based method, leverages the fact that some low-frequency components in the graph spectrum are more robust to edge perturbations and regularizes the training process of a given GCN with robust information from an auxiliary regularization network to improve the adversarial performance of GCNs. Overall, graph preprocessing tends to remove only a small fraction of edges from the adjacency matrix. Thus, the existing defense techniques often have high latency and do not scale to large graphs. We instead aim to improve the robustness of sparse GNNs with sparse adjacency matrices to achieve computation efficiency. As robustness generally requires more non-zero parameters, yielding parameter-efficient robust GLTs remains a challenge.

## 3 METHODOLOGY

**Notations.** Let $\mathcal{G} = \{\mathcal{V}, \mathcal{E}\}$ represent an undirected graph with $|\mathcal{V}|$ nodes and $|\mathcal{E}|$ edges. The topology of the graph can be represented with an adjacency matrix $\boldsymbol{A} \in \mathbb{R}^{|\mathcal{V}| \times |\mathcal{V}|}$, where $\boldsymbol{A}_{ij} = 1$ if there is an edge $e_{i,j} \in \mathcal{E}$ between nodes $v_i$ and $v_j$ while $\boldsymbol{A}_{ij} = 0$ otherwise. Each node $v_i \in \mathcal{V}$ has an attribute feature vector $\boldsymbol{x}_i \in \mathbb{R}^F$, where $F$ is the number of node features. Let $\boldsymbol{X} \in \mathbb{R}^{|\mathcal{V}| \times F}$ and $\boldsymbol{Y} \in \mathbb{R}^{|\mathcal{V}| \times C}$ denote the feature matrix and the labels of all nodes in the graph, respectively. In this paper, we will also represent a graph as a pair $\{\boldsymbol{A}, \boldsymbol{X}\}$. In the case of message-passing GNN, the representation of a node $v_i$ is iteratively updated by aggregating and transforming the representations of its neighbors. As an example, a two-layer(Kipf & Welling, 2016) GNN can be specified as

$$\boldsymbol{Z} = f(\{\boldsymbol{A}, \boldsymbol{X}\}, \boldsymbol{\Theta}) = \mathcal{S}(\hat{\boldsymbol{A}} \sigma(\hat{\boldsymbol{A}} \boldsymbol{X} \boldsymbol{W}_{(0)}) \boldsymbol{W}_{(1)}), \tag{1}$$

where $\boldsymbol{Z}$ is the prediction, $\boldsymbol{\Theta} = (\boldsymbol{W}_0, \boldsymbol{W}_1)$ are the weights, $\sigma(.)$ is the activation function, e.g., a rectified linear unit (ReLU), $\mathcal{S}(.)$ is the softmax function, $\hat{\boldsymbol{A}} = \tilde{\boldsymbol{D}}^{-\frac{1}{2}}(\boldsymbol{A}+\boldsymbol{I})\tilde{\boldsymbol{D}}^{-\frac{1}{2}}$ is the normalized adjacency matrix with self-loops, and $\tilde{\boldsymbol{D}}$ is the degree matrix of $\boldsymbol{A}+\boldsymbol{I}$. We consider the transductive semi-supervised node classification (SSNC) task for which the cross-entropy (CE) loss over labeled nodes is given by

$$\mathcal{L}_0(f(\{\boldsymbol{A}, \boldsymbol{X}\}, \boldsymbol{\Theta})) = -\sum_{l \in \mathcal{Y}_{TL}} \sum_{j=1}^{C} \boldsymbol{Y}_{l_j} \log(\boldsymbol{Z}_{l_j}), \tag{2}$$

where $\mathcal{Y}_{TL}$ is the set of train node indices, $C$ is the total number of classes, and $\boldsymbol{Y}_l$ is the one-hot encoded label of node $v_l$.

**Graph Lottery Tickets**. A GLT consists of a sparsified graph, obtained by pruning some edges in $\mathcal{G}$, and a GNN sub-network, with the original initialization, that can be retrained to achieve comparable performance to the original GNN trained on the full graph, where performance is measured in terms of test accuracy. Given a GNN $f(\cdot, \boldsymbol{\Theta})$ and a graph $\mathcal{G} = \{\boldsymbol{A}, \boldsymbol{X}\}$, the associated GNN sub-network and the sparsified graph can be represented as $f(\cdot, \boldsymbol{m}_\theta \odot \boldsymbol{\Theta})$ and $\mathcal{G}_s = \{\boldsymbol{m}_g \odot \boldsymbol{A}, \boldsymbol{X}\}$, respectively, where $\boldsymbol{m}_g$ and $\boldsymbol{m}_\theta$ are differentiable masks applied to the adjacency matrix $\boldsymbol{A}$ and the model weights $\boldsymbol{\Theta}$, respectively, and $\odot$ is the element-wise product. UGS (Chen et al., 2021) finds the two masks $\boldsymbol{m}_g$ and $\boldsymbol{m}_\theta$ such that the sub-network $f(\cdot, \boldsymbol{m}_\theta \odot \boldsymbol{\Theta})$ along with the sparsified graph $\mathcal{G}_s$ can be trained to a similar accuracy as $f(., \boldsymbol{\Theta})$.

**Poisoning Attack on Graphs.** In this work, we investigate the robustness of GLTs under non-targeted poisoning attacks modifying the structure of the graph. In the case of a poisoning attack, GNNs are trained on a graph that attackers maliciously modify. The aim of the attacker is to find an optimal perturbed $\boldsymbol{A}'$ that fools the GNN into making incorrect predictions. This can be formulated as a bi-level optimization problem (Zügner et al., 2018; Zügner & Günnemann, 2019):

$$\arg\max_{\boldsymbol{A}' \in \Phi(\boldsymbol{A})} \mathcal{L}_{atk}(f(\{\boldsymbol{A}', \boldsymbol{X}\}, \boldsymbol{\Theta}^*))$$
$$\text{s.t.} \quad \boldsymbol{\Theta}^* = \arg\min_{\boldsymbol{\Theta}} \mathcal{L}_0(f(\{\boldsymbol{A}', \boldsymbol{X}\}, \boldsymbol{\Theta})) \tag{3}$$

where $\Phi(\boldsymbol{A})$ is the set of adjacency matrices that fit the constraint $\frac{||\boldsymbol{A}'-\boldsymbol{A}||_0}{||\boldsymbol{A}||_0} \leq \Delta$, $\mathcal{L}_{atk}$ is the attack loss function, $\Delta$ is the perturbation rate, and $\boldsymbol{\Theta}^*$ is the optimal parameter for the GNN on the perturbed graph.

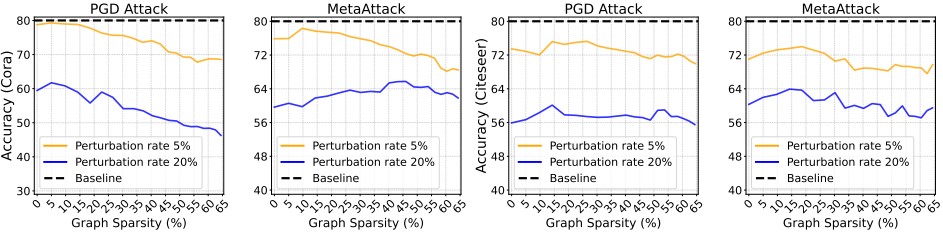

Figure 2: Classification accuracy of GLTs generated using UGS for Cora and citeseer datasets attacked by PGD and MetaAttack. The baseline refers to accuracy on the clean graph.

### 3.1 UGS ANALYSIS UNDER ADVERSARIAL ATTACKS

We perform MetaAttack (Zügner & Günnemann, 2019) and the PGD attack (Wu et al., 2019) on the Cora and citeseer datasets with different perturbation rates. We use the same setup as (Xu et al., 2018; Zhang & Zitnik, 2020; Mujkanovic et al., 2022) for performing the poisoning attacks on the datasets. Then, we apply UGS on these perturbed graphs to find the GLTs. As shown in Figure 2, the classification accuracy of the GLTs identified by UGS is lower than the clean graph accuracy. The difference increases substantially when the perturbation rate increases. For example, in the PGD attack, when the graph sparsity is 30%, at 5% perturbation, the accuracy drop is 6%. This drop increases to 25% when the perturbation rate is 20%. Moreover, for 20% perturbation rate, even with 0% sparsity, the accuracy of the GNN is around 20% lower than that of the clean graph accuracy. While UGS removes edges from the perturbed adjacency matrix, as shown in Figure 2, it

may not effectively remove the adversarially perturbed edges. A naïve application of UGS may not be sufficient to improve the adversarial robustness of the GLTs. Consequently, there is a need for an adversarially robust UGS technique that can efficiently remove the edges affected via adversarial perturbations while pruning the adjacency matrix and the associated GNN, along with improved adversarial training, allowing the dual benefits of improved robustness and inference latency.

## 3.2 Analyzing the Impact of Adversarial Attacks on Graph Properties

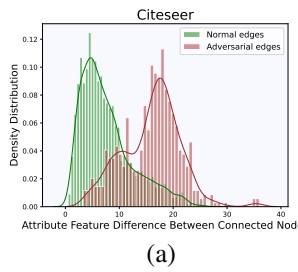 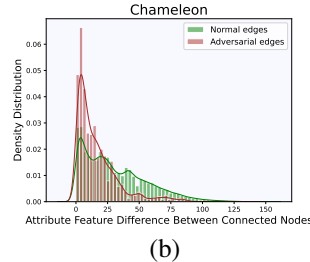 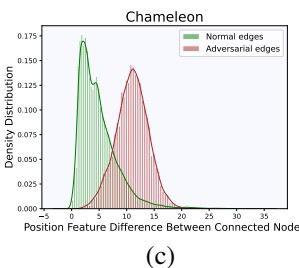

(a)            (b)            (c)

Figure 3: Impact of adversarial attacks on graph properties. (a), (b) Density distribution of attribute feature differences of connected nodes in perturbed homophilic (Citeseer) and heterophilic (Chameleon) graphs. (c) Density distribution of positional feature differences of connected nodes in perturbed heterophilic graphs.

Adversarial attacks like MetaAttack, PGD, and PR-BCD poison the graph structure by either introducing new edges or deleting existing edges, resulting in changes in the original graph properties. We analyze the difference in the attribute features of the nodes that are connected by the clean and adversarial edges. Figure 3a and b depict the density distribution of the attribute feature difference of connected nodes in homophilic and heterophilic graph datasets attacked by the PGD attack. In homophilic graphs, the attack tends to connect nodes with large attribute feature difference. A defense technique can potentially leverage this information to differentiate between the benign and adversarial edges in the graph. However, this is not the case for heterophilic graphs. We resort, instead, to the positional features of the nodes, using positional encoding techniques like DeepWalk (Perozzi et al., 2014). Interestingly, as we observe from Figure 3c, in heterophilic graphs, attacks tend to connect nodes with large positional feature difference. ARGS uses these graph properties to iteratively prune the adversarial edges from homophilic and heterophilic graphs.

## 3.3 Adversarially Robust Graph Sparsification

We present ARGS, a sparsification technique that simultaneously reduces edges in $\mathcal{G}$ and GNN parameters in $\Theta$ under adversarial attack conditions to effectively accelerate GNN inference yet maintain robust classification accuracy. ARGS reformulates the loss function to include (a) a CE loss term on the train nodes, (b) a CE loss term on a set of test nodes, and (c) a square loss term on all edges. Pruning the edges based on this combined loss function results in the removal of adversarial as well as less-important non-adversarial edges from the graph.

*Removing Edges Around Train Nodes.* Poisoning attacks like the MetaAttack and the PGD attack tend to modify more the local structure around the train nodes than that around the test nodes (Li et al., 2023). Specifically, a large portion of the modifications is introduced to the edges connecting a train node to a test node or a train node to another train node. We include a CE loss term associated with the train nodes, as defined in equation 2 in our objective function to account for the edges surrounding the train nodes. These edges include both adversarial and non-adversarial edges.

*Removing Adversarial Edges.* In numerous application domains, including social graphs, web page graphs, and citation graphs, connected nodes in a homophilic graph exhibit similar attribute features, while they still keep similar positional features in heterophilic graphs (Li et al., 2022; McPherson et al., 2001; Kipf & Welling, 2016). On the other hand, as shown in Figure 3, adversarial attacks tend to connect nodes with distinct attribute features in homophilic graphs and distinct positional features in heterophilic graphs. Therefore, we help remove adversarial edges and encourage feature smoothness by including the following loss to our objective function for homophilic graphs:

$$\mathcal{L}_{fs}(\boldsymbol{A}^{'}, \boldsymbol{X}) = \frac{1}{2} \sum_{i,j=1} \boldsymbol{A}^{'}_{ij}(\boldsymbol{x_i} - \boldsymbol{x_j})^2, \tag{4}$$

where $\boldsymbol{A}'$ is the perturbed adjacency matrix and $(\boldsymbol{x}_i - \boldsymbol{x}_j)^2$ measures the attribute feature difference. For heterophilic graphs, we introduce instead the following loss term:

$$\mathcal{L}_{fs}(\boldsymbol{A}') = \frac{1}{2} \sum_{i,j=1} \boldsymbol{A}'_{ij}(\boldsymbol{y_i} - \boldsymbol{y_j})^2, \tag{5}$$

where $\boldsymbol{y}_i, \boldsymbol{y}_j \in \mathbb{R}^P$ are the positional features of nodes $i, j$, obtained by running the DeepWalk algorithm (Perozzi et al., 2014) on the input graph $\mathcal{G}$, $P$ is the number of node positional features, and $(\boldsymbol{y}_i - \boldsymbol{y}_j)^2$ measures the positional feature difference.

*Removing Edges Around Test Nodes.* Removal of edges tends to be random in later iterations of UGS (Hui et al., 2023) since only a fraction of edges in $\mathcal{G}$ is related to the train nodes and directly impacts the corresponding CE loss. To better guide the edge removal around the test nodes, we also introduce a CE loss term for these nodes. However, the labels of the test nodes are unknown. We can then leverage the fact that structure poisoning attacks modify only the structure surrounding the train nodes, while their features and labels remain clean. Therefore, we first train a simple multi-layer perceptron (MLP) with 2 layers on the train nodes. MLPs only use the node features for training. We then use the trained MLP to predict the labels for the test nodes. We call these labels *pseudo labels*. Finally, we use the test nodes for which the MLP has high prediction confidence for computing the test node CE loss term. Let $\mathcal{Y}_{PL}$ be the set of test nodes for which the MLP

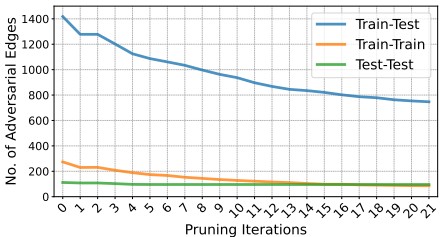

Figure 4: Evolution of adversarial edges in Cora dataset (attacked by PGD, 20% perturbation) as we apply ARGS to prune the graph. Train-Train edges connect two nodes from the train set. Train-Test edges connect a node from train set with one from test set. Test-Test edges connect two nodes from the test set.

prediction confidence is high and $\boldsymbol{Y}_{mlp}$ be the prediction by the MLP. The CE loss is given by

$$\mathcal{L}_1(f(\{\boldsymbol{A}', \boldsymbol{X}\}, \boldsymbol{\Theta})) = - \sum_{l \in \mathcal{Y}_{TL}} \sum_{j=1}^{C} \boldsymbol{Y}_{mlp_{l_j}} \log(\boldsymbol{Z}_{l_j}). \tag{6}$$

In summary, the complete loss function that ARGS optimizes is

$$\mathcal{L}_{ARGS} = \alpha \mathcal{L}_0(f(\{\boldsymbol{m}_g \odot \boldsymbol{A}', \boldsymbol{X}\}, \boldsymbol{m}_\theta \odot \boldsymbol{\Theta})) + \beta \mathcal{L}_{fs}(\boldsymbol{m}_g \odot \boldsymbol{A}', \boldsymbol{X})$$
$$+ \gamma \mathcal{L}_1(f(\{\boldsymbol{m}_g \odot \boldsymbol{A}', \boldsymbol{X}\}, \boldsymbol{m}_\theta \odot \boldsymbol{\Theta})) + \lambda_1 ||\boldsymbol{m}_g||_1 + \lambda_2 ||\boldsymbol{m}_\theta||_1, \tag{7}$$

where $\beta, \lambda_1,$ and $\lambda_2$ are the hyperparameters and the value of $\alpha$ and $\gamma$ is set to 1. $\lambda_1$ and $\lambda_2$ are the $l_1$ regularizers of $\boldsymbol{m}_g$ and $\boldsymbol{m}_\theta$, respectively. After the training is complete, the lowest percentages $p_g$ of elements of $\boldsymbol{m}_g$ and $p_\theta$ of elements of $\boldsymbol{m}_\theta$ are set to 0. Then, the updated masks are applied to prune $\boldsymbol{A}$ and $\boldsymbol{\Theta}$, and the weights of the GNN are rewound to their original initialization value to generate the ARGLT. We apply these steps iteratively until we reach the desired sparsity $s_g$ and $s_\theta$. Algorithm 1 illustrates our iterative pruning process. $|| \cdot ||_0$ is the $L_0$ norm, counting the number of non-zero elements. As shown in Figure 4 for the Cora dataset attacked by the PGD attack with 20% perturbation, most of the adversarial perturbation edges are between train and test nodes (Li et al., 2023). Moreover, our proposed sparsification technique successfully removes many of the adversarial edges. In particular, after applying our technique for 20 iterations, where each iteration removes 5% of the graph edges, the number of train-train, train-test, and test-test adversarial edges reduces by 68.13%, 47.3%, and 14.3%, respectively.

*Training Sparse ARGLTs.* Structure poisoning attacks do not modify the labels of the nodes and the locality structure of the test nodes is less contaminated (Li et al., 2023), implying that the train node labels and the local structure of the test nodes contain relatively "clean" information. We leverage this insight and train the GNN sub-network using both train nodes and test nodes. We use a CE loss term for both the train ($\mathcal{L}_0$) and test ($\mathcal{L}_1$) nodes. Since the true labels of the test nodes are not available, we train an MLP on the train nodes and then use it to predict the labels for the test nodes (Li et al., 2018; 2023). To compute the CE loss, we use only those test nodes for which the MLP has high prediction confidence. The loss function used for training the sparse GNN on the sparse adjacency matrix generated by ARGS is

$$\min_{\boldsymbol{\Theta}} \quad \eta \mathcal{L}_0(f(\{\boldsymbol{m}_g \odot \boldsymbol{A}', \boldsymbol{X}\}, \boldsymbol{m}_\theta \odot \boldsymbol{\Theta})) + \zeta \mathcal{L}_1(f(\{\boldsymbol{m}_g \odot \boldsymbol{A}', \boldsymbol{X}\}, \boldsymbol{m}_\theta \odot \boldsymbol{\Theta})) \tag{8}$$

---

**Algorithm 1** Adversarially Robust Graph Sparsification

---

**Input**: Graph $\mathcal{G} = \{A, X\}$, GNN $f(\mathcal{G}, \Theta_0)$ with initialization $\Theta^0$, Sparsity levels $s_g$ for graph and $s_\theta$ for GNN, Initial masks $m_g = A, m_\theta = 1 \in \mathbb{R}^{||\Theta_0||_0}$
**Output**: Final masks $m_g, m_\theta$

1: **while** $\left(1 - \frac{||m_g||_0}{||A||_0} < s_g\right)$ and $\left(1 - \frac{||m_\theta||_0}{||\Theta||_0} < s_\theta\right)$ **do**
2:      $m_g^{(0)} = m_g, m_\theta^{(0)} = m_\theta, \Theta^{(0)} = \{W_0^{(0)}, W_1^{(0)}\}$
3:      **for** $t = 0, 1, 2, \ldots, T - 1$ **do**
4:          Forward $f(\cdot, m_\theta^t \odot \Theta^t)$ with $\mathcal{G} = m_g^t \odot A, X$ to compute the loss $\mathcal{L}_{ARGS}$ in equation 7
5:          $\Theta^{t+1} \leftarrow \Theta^t - \mu \nabla_{\Theta^t} \mathcal{L}_{ARGS}$
6:          $m_g^{t+1} \leftarrow m_g^t - \omega_g \nabla_{m_g^t} \mathcal{L}_{ARGS}$
7:          $m_\theta^{t+1} \leftarrow m_\theta^t - \omega_\theta \nabla_{m_\theta^t} \mathcal{L}_{ARGS}$
8:      $m_g = m_g^{T-1}, m_\theta = m_\theta^{T-1}$
9:      Set percentage $p_g$ of the lowest-scored values in $m_g$ to 0 and set others to 1
10:     Set percentage $p_\theta$ of the lowest-scored values in $m_\theta$ to 0 and set others to 1

---

where $m_\theta$ and $m_g$ are the masks evaluated by ARGS that are kept fixed throughout training, and $\eta$ and $\zeta$ are set to 1. In the early pruning iterations, when graph sparsity is low, the test nodes are more useful in improving the model adversarial performance because the train nodes' localities are adversarially perturbed and there exist distribution shifts between the train and test nodes. However, as the graph sparsity increases, adversarial edges associated with the train nodes are gradually removed by ARGS, thus reducing the distribution shift and making the contribution of the remaining train nodes more important in the adversarial training.

## 4 EVALUATION

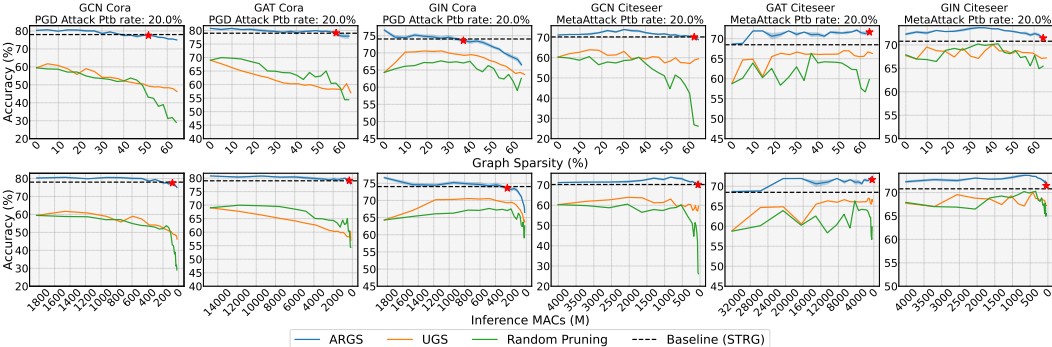

Figure 5: Node classification performance versus graph sparsity levels and inference MACs for the GCN, GIN, and GAT architectures on Cora and Citeseer datasets attacked by PGD and MetaAttack, respectively. Red stars ⋆ indicate the ARGLTs. Dash black lines represent the performance of STRG, an adversarial defense technique.

**Evaluation Setup.** We evaluate the effectiveness of ARGS and the existence of ARGLTs across diverse datasets and GNN models under different adversarial attacks and perturbation rates. In particular, we evaluate our sparsification method on both homophilic and heterophilic graph datasets which are attacked by two structure poisoning attacks, namely, PGD (Wu et al., 2019) and MetaAttack (Zügner & Günnemann, 2019). We also evaluate ARGS on a large dataset, namely, OGBN-ArXiv (Hu et al., 2020), attacked by the PR-BCD attack. Finally, we evaluate the robustness of ARGS against adaptive attacks (Mujkanovic et al., 2022). We compare our method with UGS (Chen et al., 2021), random pruning, and other state-of-the-art adversarial defense methods, namely, STRG (Li et al., 2023), GARNET (Deng et al., 2022), GNNGuard (Zhang & Zitnik, 2020), ProGNN (Jin et al., 2020a), and Soft Median (Geisler et al., 2021). Only UGS and random pruning techniques prune both the graph adjacency matrix and the GNN model parameters – no other existing defense techniques prune the GNN model parameters. For a fair comparison, we set $p_g = 5\%, p_\theta = 20\%$, similarly to the parameters used by UGS. More details on the dataset statistics, model configurations, and hyperparameters in ARGS can be found in appendix A.1.

**Defense on Homophilic Graphs.** We first evaluate the performance of ARGS on homophilic graphs against PGD and MetaAttack. Due to space limitations, we show the results for a 20% perturbation rate for the Cora and citeseer datasets. Results for PubMed and other perturbation rates

are shown in Appendix A.3. We consider 3 different backbones, namely, graph convolution networks (GCNs) (Kipf & Welling, 2016), graph isomorphism networks (GINs) (Xu et al., 2018), and graph attention networks (Veličković et al., 2017). Figure 5 shows the results for the GCN, GIN, and GAT architectures on the Cora and Citeseer datasets attacked by PGD and MetaAttack, respectively, where the average accuracy of ARGLT is reported across 5 runs. ARGLTs at a range of graph sparsity from 30% to 60% with similar performance as the STRG baseline can be identified across the different GNN backbones. The ARGLTs significantly reduce the inference MACs for GCN, GIN, and GAT by 95%, 97%, and 83%, respectively, for the Cora dataset. For the citeseer dataset, the inference MACs reduce by 98% for all the backbone GNNs.

**Comparison with Other Defense Techniques.** We compare the performance of ARGS with GNNGuard, GARNET, and ProGNN, which are all defense methods. Differently from ARGS, none of these methods prunes the weights of the GNN model. We compare these methods in terms of accuracy and inference MAC and we consider GCN as the backbone. For the different baselines, the GLT which has similar accuracy as the baseline with maximum graph and model sparsity is identified as the ARGLT by ARGS. As evident in Table 1, compared to these techniques, ARGS is always able to find GLTs that have better or similar classification accuracy with a much lower number of inference MACs.

Table 1: Performance comparison between ARGS and other defense techniques in terms of accuracy and inference MAC count.

| | Cora (PGD attack) Perturbation Rate 20% | | Citeseer (MetaAttack) Perturbation Rate 20% | |
|---|---|---|---|---|
| Model | Accuracy (%) | Inference MACs (M) | Accuracy (%) | Inference MACs (M) |
| GCN-ProGNN | 63.43±0.89 | 1832.14 | 61.02 ±0.11 | 4006.91 |
| GCN-ARGS | **77.53±1.15** | **78.78** | **70.2 ±0.89** | **252.01** |
| GCN-GNNGuard | 73.19±0.72 | 1948.32 | 71.62±1.01 | 4188.33 |
| GCN-ARGS | **77.53±1.15** | **78.78** | **71.78±0.58** | **333.83** |
| GCN-GARNET | 66.66±1.10 | 1684.9 | 72.97±1.20 | 3898.21 |
| GCN-ARGS | **77.53±1.15** | **78.78** | **73.19±0.78** | **400.6** |

**Defense on Large Graphs.** We evaluate the robustness of ARGS on the large-scale dataset OGBN-ArXiv. We use the PR-BCD attack for perturbing the dataset and the reference GNN model is GCN. PGD or MetaAttack faces timeout due to memory for these large graphs. GCN is used as the baseline. Figure 6 shows that ARGS is able to identify ARGLTs that have high model and graph sparsity. In particular, the model sparsity and graph sparsity are 79.03% and 30.17% for the 10% perturbed dataset, and 78.31% and 30.81%, respectively, for the 15% perturbed dataset. The inference MACs reduce by 79% and 78% for the 10%-and 15%-perturbed OGBN-ArXiv dataset. These results show that ARGS is able to find highly sparse GLTs also for large-scale datasets.

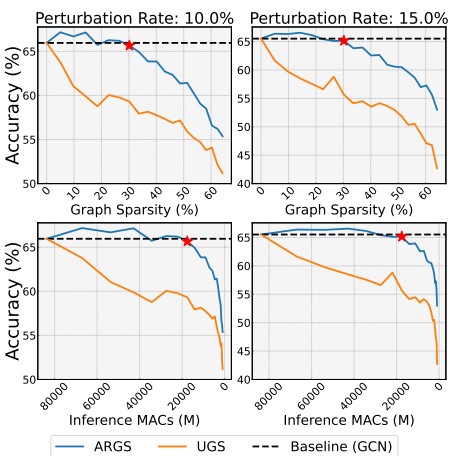

Figure 6: Node classification performance versus graph sparsity levels and inference MACs for GCN on OGBN-ArXiv dataset attacked by PR-BCD.

**Defense Against Adaptive Attacks.** Recently, adaptive attacks (Mujkanovic et al., 2022) have been developed, which are targeted to specific defense techniques, to evaluate their robustness. Because all the components in the loss function of ARGS are differentiable, ARGS can also be directly attacked by an adaptive attack. Specifically, we evaluate ARGS on a gradient-based adaptive attack, called Meta-PGD (Mujkanovic et al., 2022), where ARGS is attacked by unrolling its training procedure. Table 2 compares the performance of ARGS against the PGD attack and the adaptive attack, with GCN as the GNN backbone for Cora. For a 5% perturbation rate, the accuracy of the ARGLT identified by ARGS reduces by only ∼ 1.7%. For a 10% perturbation rate, the reduction in classification accuracy is ∼ 2.9% for the Cora dataset, showing that ARGS is also robust to adaptive attacks. Table 2 shows that the classification accuracy of the ARGLTs is on average 9% more than the GLTs identified by UGS. We include the results for Citeseer in A.5.

Table 2: ARGS and UGS performance comparison for the PGD attack and an adaptive attack for the Cora dataset. GCN is used as the GNN model.

| Dataset | Technique | Attack | Classification Accuracy at Perturbation Rate 5% | | Classification Accuracy at Perturbation Rate 10% | |
| | | | Graph Sparsity 22.64% Model Sparsity 67.60% | Graph Sparsity 43.16% Model Sparsity 91.70% | Graph Sparsity 22.64% Model Sparsity 67.60% | Graph Sparsity 43.16% Model Sparsity 91.70% |
|---|---|---|---|---|---|---|
| Cora | UGS | PGD Attack | 75.95±0.19 | 72.11±0.40 | 69.48±0.28 | 66.18±0.13 |
| | | Adaptive Attack | 74.13±0.20 | 70.98±0.33 | 68.02±0.45 | 64.79±0.34 |
| Cora | ARGS | PGD Attack | **82.04±1.09** | **80.68±0.85** | **82.8±0.77** | **80.18±1.13** |
| | | Adaptive Attack | **80.33±1.35** | **78.77±1.86** | **79.68±1.35** | **77.16±0.98** |

**Defense on Heterophilic Graphs.** We report the classification accuracy of ARGS on heterophilic graphs in Figure 7. We use GPRGNN (Chien et al., 2020) as the GNN model for heterophilic graphs and the Chameleon and Squirrel (McCallum et al., 2000) datasets. GPRGNN performs better than GCN, GIN, and GAT for heterophilic graphs (Deng et al., 2022). We use GARNET as the baseline since it achieves state-of-the-art adversarial classification accuracy compared to other defense techniques for heterophilic graphs. As shown in Figure 7, ARGS is able to identify GLTs that achieve similar classification accuracy as GARNET for the Chameleon and Squirrel datasets attacked by PGD and MetaAttack with 85% to 97% weight sparsity, resulting in a substantial reduction in inference MACs.

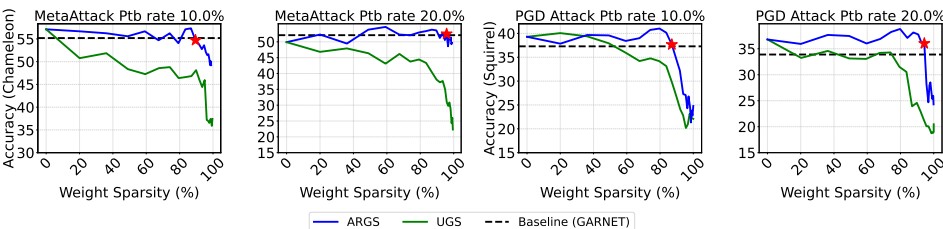

Figure 7: Node classification performance versus weight sparsity levels for GPRGNN on Chameleon and Squirrel dataset attacked by the PGD Attack and MetaAttack.

**Ablation Study.** We evaluate the effectiveness of each component of the proposed loss function for the sparsification algorithm by performing an ablation study, as shown in Table 3. We consider the Cora dataset under PGD attack with 10% and 20% perturbation rates. Configuration 1 corresponds to ARGS with all the loss components (equation 7). Configuration 2 does not use the feature smoothness component (equation 4) while configuration 3 skips the CE loss associated with the predicted test nodes (equation 6). Configuration 4 skips both the smoothness and CE loss on predicted test nodes. Table 3 shows that both configurations 2 and 3 improve the final performance when compared to that of configuration 4, highlighting the importance of the losses introduced in equation 4 and equation 6. More importantly, at both high and low target sparsity, we yield the best classification performance with configuration 1, showcasing the importance of the unified loss function (equation 7). Further ablation studies are provided in the appendix.

Table 3: Ablation study.

| GCN, Cora, PGD Attack | | | | | | Classification Accuracy at Perturbation Rate 10% | | Classification Accuracy at Perturbation Rate 20% | |
| Configuration | $\alpha$ | $\beta$ | $\gamma$ | $\eta$ | $\zeta$ | Graph Sparsity 9.8% Model Sparsity 36.1% | Graph Sparsity 64.4% Model Sparsity 98.9% | Graph Sparsity 9.8% Model Sparsity 36.1% | Graph Sparsity 64.5% Model Sparsity 98.9% |
|---|---|---|---|---|---|---|---|---|---|
| 1 | ✓ | ✓ | ✓ | ✓ | ✓ | **83.25** | **75.10** | **80.63** | **75.60** |
| 2 | ✓ | ✗ | ✓ | ✓ | ✓ | 82.04 | 70.57 | 78.92 | 64.84 |
| 3 | ✓ | ✓ | ✗ | ✓ | ✓ | 82.44 | 72.84 | 78.97 | 52.92 |
| 4 | ✓ | ✗ | ✗ | ✓ | ✓ | 80.58 | 62.42 | 75.7 | 54.18 |

## 5 CONCLUSION

In this paper, we first empirically observed that the performance of GLTs collapses against structure perturbation poisoning attacks. To address this issue, we presented a new adversarially robust graph sparsification technique, ARGS, that prunes the perturbed adjacency matrix and the GNN weights by optimizing a novel loss function. By iteratively applying ARGS, we found ARGLTs that are highly sparse yet achieve competitive performance under different structure poisoning attacks. Our evaluation showed the superiority of our method over UGS at both high and low sparsity regimes.

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

## A APPENDIX

### A.1 DATASET DETAILS

We use seven different benchmark datasets, namely, Cora, Citeseer, PubMed, OGBN-arXiv, Chameleon, and Squirrel, to evaluate the efficacy of ARGS in finding ARGLTs. Details about the benchmark datasets are summarized in Table 4. We consider the largest connected component (LCC) for each of these datasets.

Table 4: Details on the datasets.

| Datasets | Type | #Nodes | #Edges | Classes | Features |
|---|---|---|---|---|---|
| Cora | Homophilic | 2485 | 5069 | 7 | 1433 |
| Citeseer | Homophilic | 2110 | 3668 | 6 | 3703 |
| PubMed | Homophilic | 19717 | 44338 | 3 | 500 |
| OGBN-ArXiv | Homophilic | 169,343 | 1,166243 | 40 | 128 |
| OGBN-Products | Homophilic | 2,449,029 | 61,859,140 | 47 | 100 |
| Chameleon | Heterophilic | 2277 | 62792 | 5 | 2325 |
| Squirrel | Heterophilic | 5201 | 396846 | 5 | 2089 |

### A.2 IMPLEMENTATION DETAILS

For fair comparison, we follow the setup used by UGS as our default setting. For Cora, Citeseer, and PubMed we conduct all our experiments on two-layer GCN/GIN networks with 512 hidden units. The graph sparsity $p_g$ and model sparsity $p_\theta$ are $5\%$ and $20\%$ unless otherwise stated. The value of $\beta$ is chosen from $[0, 0.01, 0.1, 1]$ while the value of $\alpha, \gamma, \eta$, and $\zeta$ is 1 by default, respectively. We use Adam optimizer for training the GNNs. The value of $\lambda_1$ and $\lambda_2$ is $10^{-2}, 10^{-2}$ for Cora and Citeseer while for PubMed it is 1e-6, 1e-3 respectively. In each pruning round, the number of epochs to update masks is by default 200 and we use early stopping here. The 2-layer MLP used for predicting the pseudo labels of the test nodes has hidden dimension of 1024. We use DeepRobust, an adversarial attack repository (Li et al., 2020), to implement the PGD attack and MetaAttack on the different datasets. We use Pytorch-Geometric (Fey & Lenssen, 2019) for performing the PR-BCD attack on OGBN-ArXiv dataset. NVIDIA Tesla A100 (80GB GPU) is used to conduct all our experiments.

## A.3 Performance evaluation of ARGS

We evaluate the performance of ARGS on homophilic graphs perturbed by PGD and MetaAttack. We show the results of $5\%, 10\%$, and $15\%$, perturbation rate for the Cora and Citeseer datasets in Figure 8. Figure 9 shows the performance of ARGS on the Cora and Citeseer datasets with GIN as the backbone GNN. Finally, Figure 10 shows the performance of ARGS on Citeseer with GAT as the backbone GNN.

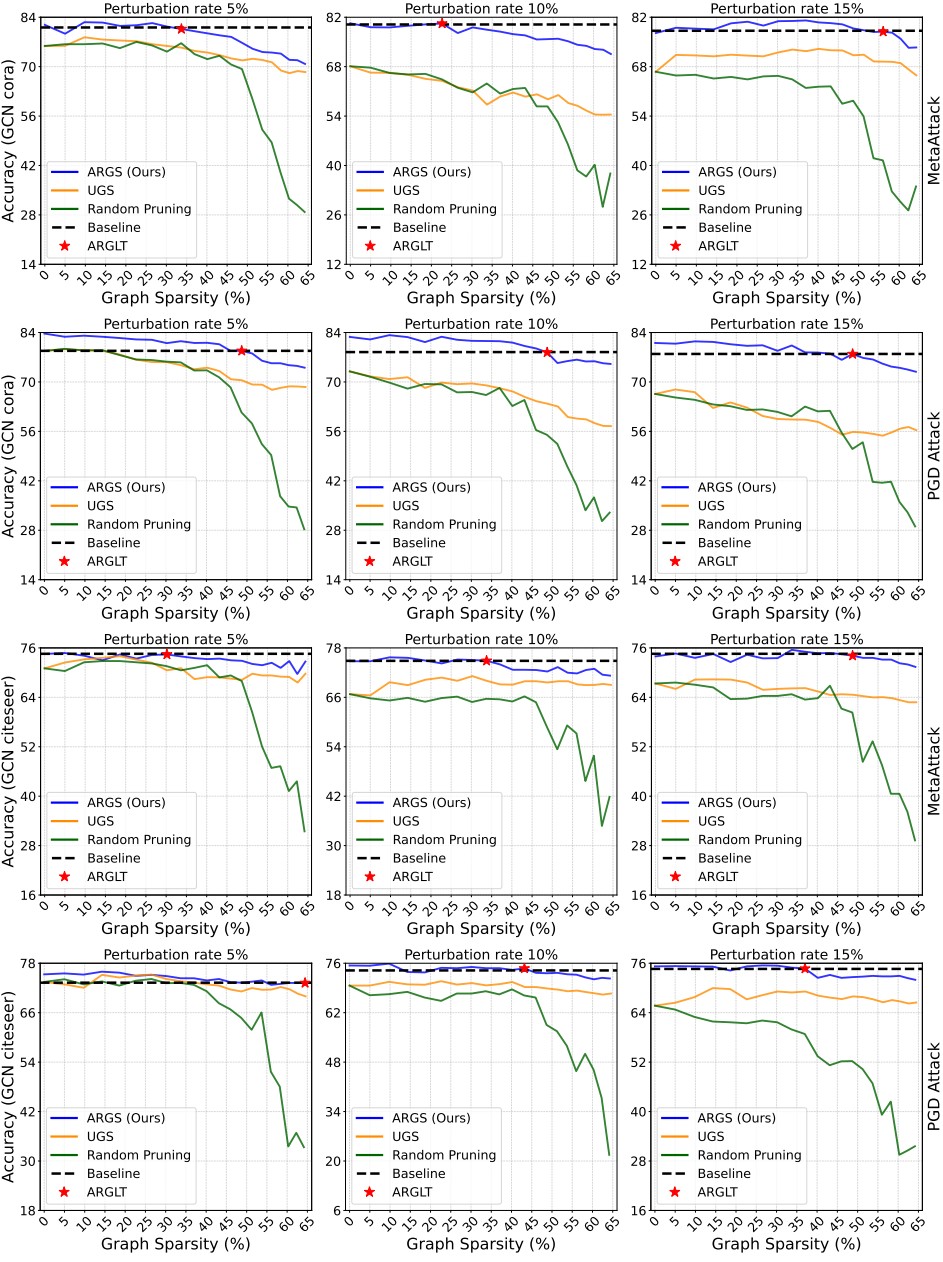

Figure 8: Node classification performance over achieved graph sparsity levels for GCN on Cora and Citeseer datasets attacked by PGD and MetaAttack. The perturbation rates are 5%, 10%, 15%, and 20%. Red stars ⋆ indicate the ARGLTs which achieve similar performance with high sparsity. STRG is used as the baseline.

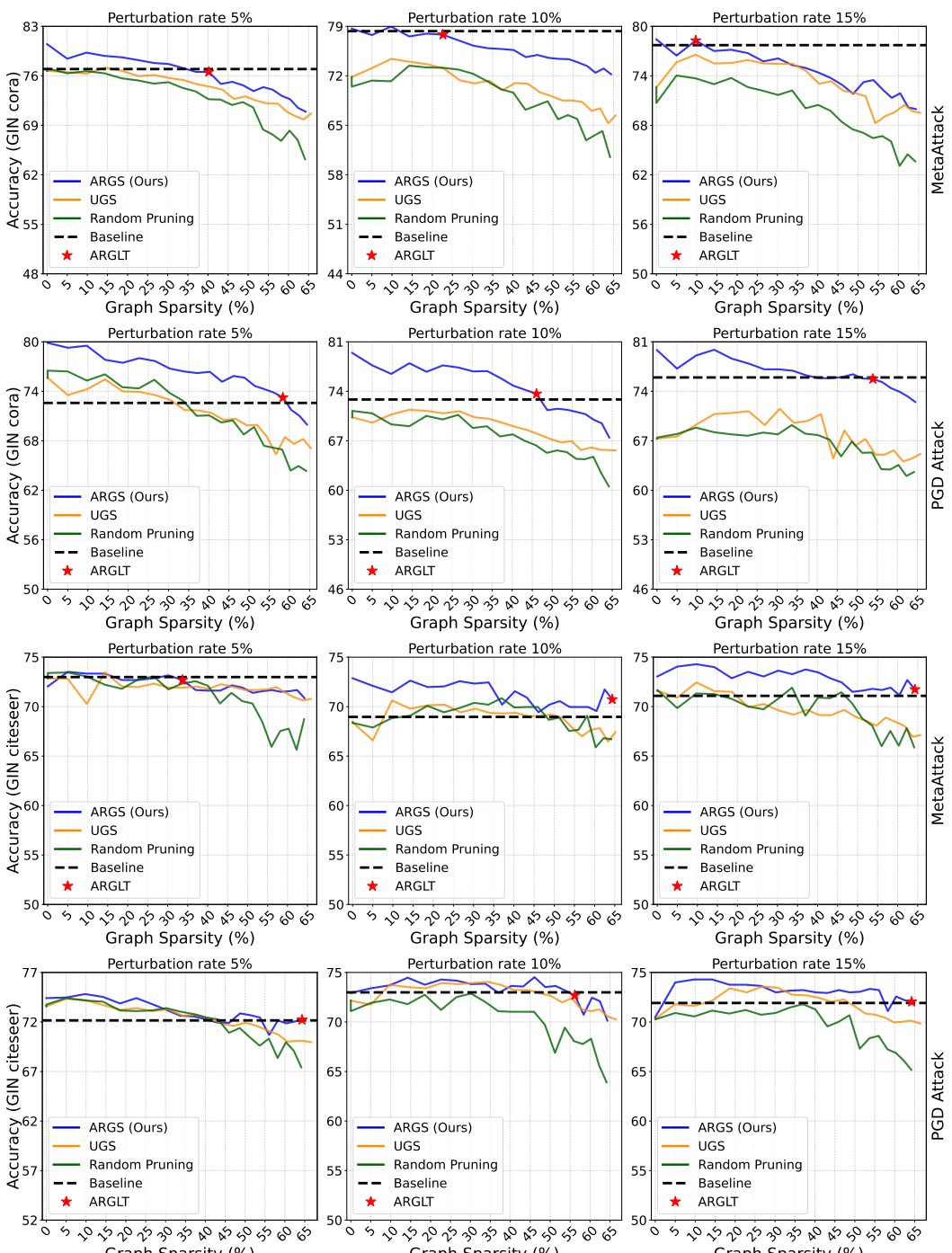

Figure 9: Node classification performance over achieved graph sparsity levels for GIN on Cora and Citeseer datasets attacked by PGD and MetaAttack. The perturbation rates are 5%, 10%, 15%, and 20%. Red stars ⋆ indicate the ARGLTs which achieve similar performance with high sparsity. STRG is used as the baseline.

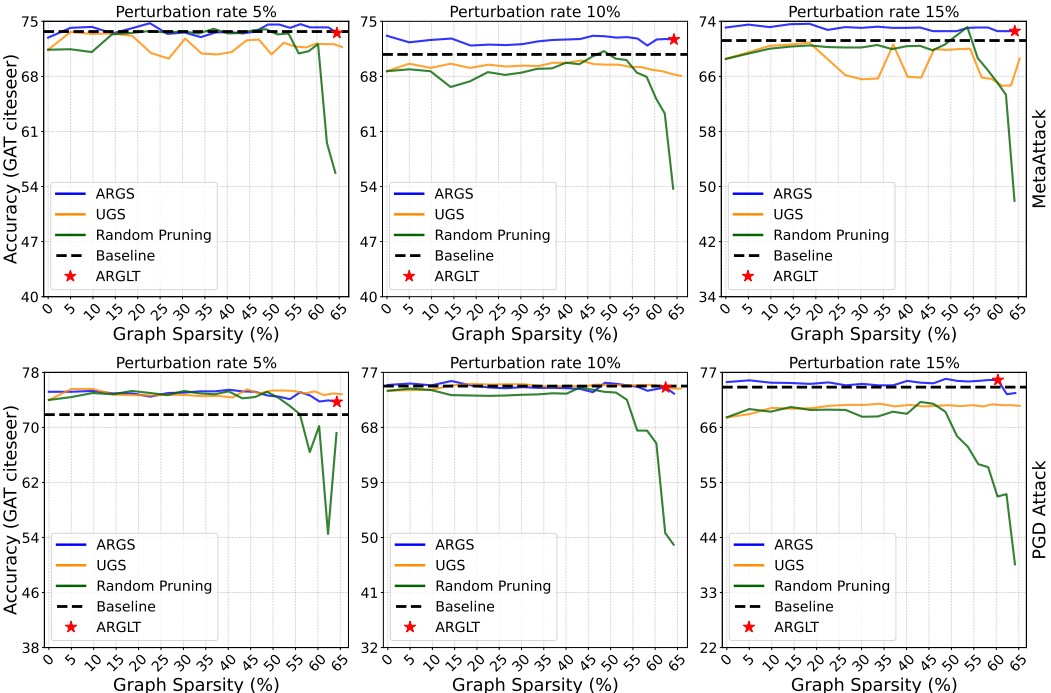

Figure 10: Node classification performance over achieved graph sparsity levels for GAT on Citeseer datasets attacked by PGD and MetaAttack. The perturbation rates are 5%, 10% and 15%. Red stars ⋆ indicate the ARGLTs which achieve similar performance with high sparsity.STRG is used as the baseline.

Furthermore, we also evaluate the performance of ARGS on the PubMed dataset and Figure 11 depicts the performance of ARGS. As evident from the figures, ARGS performs better than UGS and it is able to find highly sparse ARGLTS.

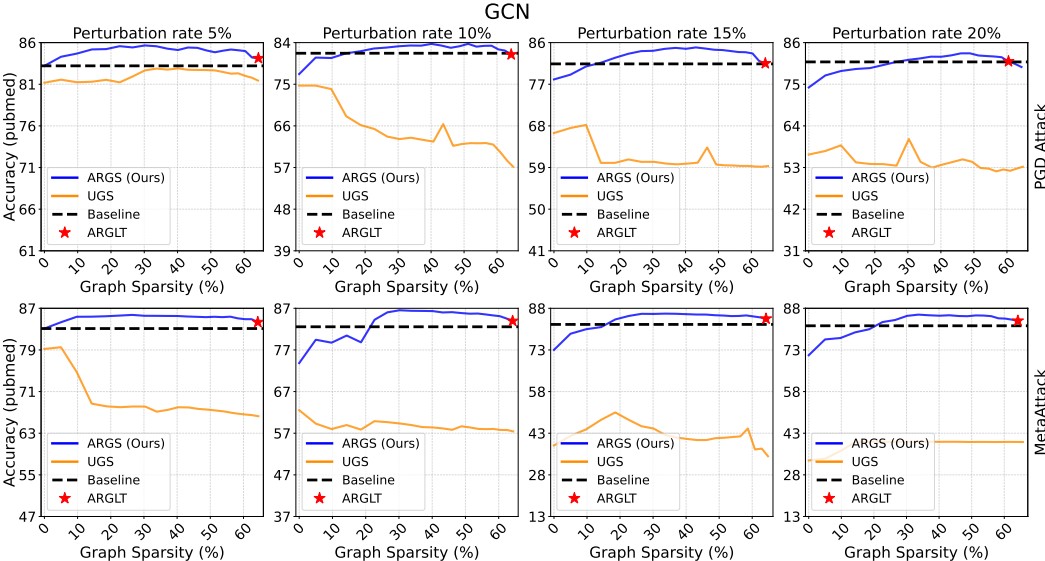

Figure 11: Node classification performance over achieved graph sparsity levels for GCN on PubMed dataset attacked by PGD and MetaAttack. The perturbation rates are 5%, 10%, 15%, and 20%. Red stars ⋆ indicate the ARGLTs which achieve similar performance with high sparsity.STRG is used as the baseline.

## A.4 ABLATION STUDY

To verify the effectiveness of each component of the proposed loss function used for the sparsification algorithm, we perform an ablation study. A part of this ablation study is already included in the original paper. We include rest of the results here in table 5. In particular, we present extensive ablation study performed on Cora dataset for all the 3 different attacks with all 4 different perturbation rates. We recall, Configuration 1 corresponds to ARGS with all the loss components. As shown in Table 5, at both high and low target sparsity, we yield the best classification performance with configuration 1, showcasing the importance of the unified loss function.

Table 5: Ablation study.

| GCN, Cora, PGD Attack | | | | | | Classification Accuracy at Perturbation Rate 5% | | Classification Accuracy at Perturbation Rate 15% | |
|---|---|---|---|---|---|---|---|---|---|
| Configuration | $\alpha$ | $\beta$ | $\gamma$ | $\eta$ | $\zeta$ | Graph Sparsity 22.7% Model Sparsity 67.7% | Graph Sparsity 60.4% Model Sparsity 98.2% | Graph Sparsity 22.7% Model Sparsity 67.7% | Graph Sparsity 60.4% Model Sparsity 98.2% |
| 1 | ✓ | ✓ | ✓ | ✓ | ✓ | **82.04** | **74.75** | **80.23** | **73.99** |
| 2 | ✓ | ✗ | ✓ | ✓ | ✓ | 81.84 | 73.64 | 79.98 | 68.81 |
| 3 | ✓ | ✓ | ✗ | ✓ | ✓ | 81.69 | 74.45 | 76.86 | 72.89 |
| 4 | ✓ | ✗ | ✗ | ✓ | ✓ | 79.28 | 71.33 | 74.70 | 63.48 |
| GCN, Cora, Mettack Attack | | | | | | Classification Accuracy at Perturbation Rate 5% | | Classification Accuracy at Perturbation Rate 10% | |
| Configuration | $\alpha$ | $\beta$ | $\gamma$ | $\eta$ | $\zeta$ | Graph Sparsity 22.7% Model Sparsity 67.6% | Graph Sparsity 62.3% Model Sparsity 98.6% | Graph Sparsity 22.6% Model Sparsity 67.5% | Graph Sparsity 64.2% Model Sparsity 98.9% |
| 1 | ✓ | ✓ | ✓ | ✓ | ✓ | **81.74** | **71.83** | **80.23** | **71.58** |
| 2 | ✓ | ✗ | ✓ | ✓ | ✓ | 80.89 | 69.91 | 78.17 | 70.98 |
| 3 | ✓ | ✓ | ✗ | ✓ | ✓ | 79.88 | 71.33 | 75.40 | 66.81 |
| 4 | ✓ | ✗ | ✗ | ✓ | ✓ | 78.89 | 69.03 | 75.40 | 60.97 |
| GCN, Cora, Mettack Attack | | | | | | Classification Accuracy at Perturbation Rate 15% | | Classification Accuracy at Perturbation Rate 20% | |
| Configuration | $\alpha$ | $\beta$ | $\gamma$ | $\eta$ | $\zeta$ | Graph Sparsity 22.7% Model Sparsity 67.6% | Graph Sparsity 60.4% Model Sparsity 98.2% | Graph Sparsity 22.6% Model Sparsity 67.5% | Graph Sparsity 60.3% Model Sparsity 98.2% |
| 1 | ✓ | ✓ | ✓ | ✓ | ✓ | **80.73** | **75.91** | **79.38** | **70.37** |
| 2 | ✓ | ✗ | ✓ | ✓ | ✓ | 80.23 | 73.69 | 78.72 | 69.97 |
| 3 | ✓ | ✓ | ✗ | ✓ | ✓ | 77.97 | 72.74 | 75.50 | 69.16 |
| 4 | ✓ | ✗ | ✗ | ✓ | ✓ | 78.42 | 72.08 | 74.09 | 68.86 |

## A.5 DEFENSE AGAINST ADAPTIVE ATTACKS

We perform a gradient-based adaptive attack called Meta-PGD attack (Mujkanovic et al., 2022), where ARGS is directly attacked by unrolling its training procedure. Table 6 compares the performance of ARGS against the PGD attack and the adaptive attack, with GCN as the GNN backbone for the Citeseer datasets. For a 5% perturbation rate, the accuracy of the ARGLT identified by ARGS reduces by only $\sim 1.5\%$. For a 10% perturbation rate, the reduction in classification accuracy is about $2.5\%$ for the Citeseer dataset, showing that ARGS can also be robust to adaptive attacks. The classification accuracy of the ARGLTs is on average 6% higher than the GLTs identified by UGS for the same sparsity levels.

Table 6: ARGS and UGS performance comparison for the PGD attack and an adaptive attack for the Citeseer dataset. GCN is used as the GNN model.

| Dataset | Technique | Attack | Classification Accuracy at Perturbation Rate 5% | | Classification Accuracy at Perturbation Rate 10% | |
|---|---|---|---|---|---|---|
| | | | Graph Sparsity 22.64% Model Sparsity 67.60% | Graph Sparsity 43.16% Model Sparsity 91.70% | Graph Sparsity 22.64% Model Sparsity 67.60% | Graph Sparsity 43.16% Model Sparsity 91.70% |
| Citeseer | UGS | PGD Attack | 74.63±0.30 | 72.29±0.36 | 70.70±0.26 | 69.34±0.49 |
| | | Adaptive Attack | 69.8±0.31 | 66.97±0.25 | 65.48±0.31 | 63.98±0.18 |
| Citeseer | ARGS | PGD Attack | **75.32±0.88** | **74.17±0.56** | **74.7±0.98** | **73.53±.1.05** |
| | | Adaptive Attack | **74.11±1.76** | **73.16±0.87** | **72.93±1.87** | **71.89±1.01** |

## A.6 DEFENSE ON LARGE GRAPHS

In our prior investigations, we assessed the resilience of ARGS using the OGBN-ArXiv dataset. Presently, we extend our evaluation to the OGBN-Products dataset, which is considerably larger, having 2.5 million nodes (14 times the count in OGBN-ArXiv) and 61 million edges (53 times the edges in OGBN-ArXiv). Attempting the PRBCD attack on this dataset resulted in out-of-memory errors. We then conducted a more scalable GRBCD attack (Geisler et al., 2021) on the OGBN-Products dataset, employing a perturbation rate of 50%. Subsequently, we applied our proposed technique to identify ARGLTs in the attacked graph and present the outcomes in Table 7. Our baselines include GCN, GNNGuard, and Soft Median GDC. Preliminary experiments indicate that even for larger graphs, ARGS demonstrates the capability to discover sparse ARGLTs.

Table 7: Graph sparsity, model sparsity, and inference MACs of ARGLTs across different baselines for the OGBN-Products dataset.

| Dataset | Attack | Baseline | ARGLT | | |
| --- | --- | --- | --- | --- | --- |
| | | | Graph Sparsity | Model Sparsity | Inference MACs |
| OGBN-products | 50% (GRBCD Attack) | GNNGuard | 30.17% | 79.03% | 131,143 M |
| | | GCN | 36.97% | 86.58% | 83,926 M |
| | | Soft Median GDC | 62.26% | 98.56% | 9,005 M |

## A.7 ANALYSIS UNDER STRUCTURAL AND NODE FEATURE ATTACKS

In addition to structural attacks, we also evaluate the performance of ARGS against an attack that modifies both the graph structure and the node features simultaneously. Mettack (Zügner & Günnemann, 2019) can be modified for this purpose. For a given perturbation budget, this attack performs a structure perturbation and a feature perturbation at each iteration, and between the two perturbations, it chooses the one that results in a higher attack loss. This iterative process is repeated until the perturbation budget is exhausted. We attack the Cora and Citeseer datasets with 5%, 10%, and 15% perturbation rates, and use the STRG defense technique as the baseline. ARGS is able to find highly sparse GLTs that achieve similar classification accuracy as the baseline for different graph datasets perturbed with different perturbation rates using this attack. For example, for a 5% perturbation rate, ARGS finds GLTs that have 53.75% graph sparsity and 96.55% model sparsity for the Cora dataset and 58.31% graph sparsity and 97.92% model sparsity for the Citeseer dataset. We also include the performance of UGS for comparison. Figure 12 shows that, for the same sparsity levels, GLTs identified by ARGS achieve much higher classification accuracy when compared to GLTs identified by UGS. We observe that the attacked graph contains more edge perturbations than feature perturbations since modifying the graph structure results in higher attack loss than modifying the node features. This result shows that ARGS can find highly sparse GLTs for graphs attacked by both structure and node feature perturbations.

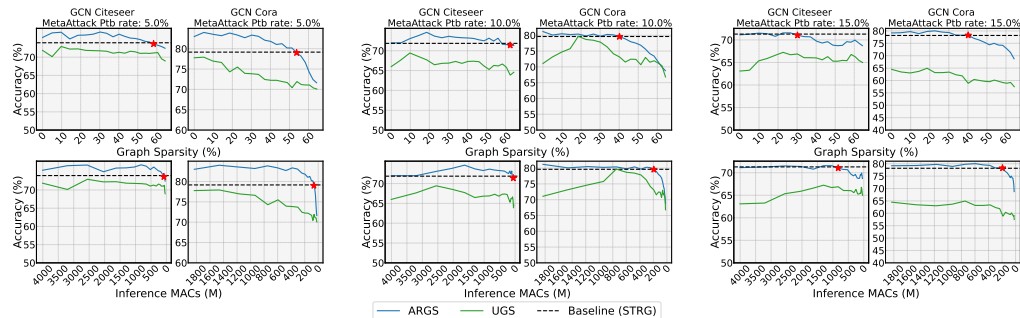

Figure 12: Node classification performance over achieved graph sparsity levels for a GCN model on Cora and Citeseer datasets attacked by a modified version of the MetaAttack that modifies both graph structure and node features. The perturbation rates are 5%, 10%, and 15%. Red stars ⋆ indicate the ARGLTs. STRG is used as the baseline.

