# OpenReview forum: "Finding Adversarially Robust Graph Lottery Tickets"
_ICLR.cc/2024/Conference — Submitted to ICLR 2024_

### Official Review · Reviewer_eZKc · 2023-10-27

**Soundness:** 2 fair
**Presentation:** 1 poor
**Contribution:** 2 fair
**Rating:** 3
**Confidence:** 5

**Summary:**

The paper proposes a novel technique for finding graph lottery tickets (GLTs), which are sparse graph neural networks (GNNs) and sparse input graphs that can reduce the inference cost and complexity compared to their dense counterparts. The technique called adversarially robust graph sparsification (ARGS), aims to improve the robustness of GLTs against different structure perturbation attacks, which are attacks that modify the graph structure to degrade the performance of GNNs. ARGS consists of pruning the perturbed adjacency matrix and the GNN weights by optimizing a novel loss function that captures the graph homophily property and the information from both the true labels and the pseudo labels. ARGS can be applied iteratively to find adversarially robust GLTs (ARGLTs) that are highly sparse yet achieve competitive performance under different structure attacks. The paper evaluates ARGS on six datasets and shows that it outperforms existing methods in terms of accuracy and sparsity.

**Strengths:**

* The paper addresses a novel and important problem of finding GLTs that are resilient to structure perturbation attacks, which can enhance the robustness and reliability of GNNs in various applications.
* The authors introduce a novel and flexible technique that can prune both the adjacency matrix and the GNN weights by using a novel loss function that incorporates both the graph homophily property and the pseudo labels.
* Extensive experiments and analysis are conducted to demonstrate the advantages of the proposed framework in the presence of adversarial examples.

**Weaknesses:**

* The novelty of the method proposed in this paper needs further clarification in my opinion, though the studied problem is novel. The method combines the existing observations into the UGS framework to handle the adversarial attack in large and densely connected graphs. However, the studied graphs are small according to [1]. Therefore, the main focus should be on large graphs such as ogb-products or ogb-papers100M to demonstrate the method’s scalability, which is not reflected in the current manuscript. Meanwhile, given the observations are intuitive and might not be generalizable in very large graphs (e.g., only observed on cora and citeseer, both have nodes less than 10k), theoretical supports that demonstrate the results could be scaled to graphs with massive nodes should be included from my perspective.

* The paper is loosely written with many grammar errors and broken format. For example, “…many defense techniques have been developed that try to improve the classification accuracy…” on Page 1. Fig. and Figure are randomly used. The same for equation and Equation. And there is a missing title for the Table on Page 8. I suggest the authors carefully revise the manuscript to make it more self-contained.

* Why the training process of ARGS (Eq.7) and the finding process of ARGLT (Eq.8) are separated but not unitedly trained as in UGS?

* Many recent works tackling graph robustness are not covered in the related works/compared as baselines, especially spectral-based methods such as [2-5].

* Experimental settings are vaguely described. For example, how the adaptive attack is performed is unclear. Meanwhile, Table 1 (which should be Table 2?) contains the results of the adaptive attack for ARGS only but no other baselines such as GNNGUARD/UGS, which could not sufficiently reflect the effectiveness of ARGS. Meanwhile, for the defense on large graphs, it is confusing whether the baseline in Figure 6 is GCN or GARNET. The baseline Soft Median from [1] should be added for better comparison on the scalability evaluation.

[1] Robustness of Graph Neural Networks at Scale, NeurIPS 2021

[2] Adversarial Attacks on Node Embeddings via Graph Poisoning, ICML 2019

[3] A Restricted Black-box Adversarial Framework Towards Attacking Graph Embedding Models, AAAI 2020

[4] Not All Low-Pass Filters are Robust in Graph Convolutional Networks, NeurIPS 2021

[5] Robust Graph Representation Learning for Local Corruption Recovery, WWW 2023

**Questions:**

Please kindly refer to the Weaknesses.

---

> ### Author Response · Authors · 2023-11-20
> **Response to reviewer eZKc (1/3)**
>
> We reference the reviewer with the identifier eZKc as R4. We thank reviewer R4 for the comments, which we address below. We thank reviewer R4 for pointing out that our paper addresses a novel and important problem statement, our proposed technique ARGS is novel and flexible, and extensive experiments demonstrate the advantage of our proposed framework. Comment n in the weaknesses is denoted by R4Wn.
>
> **R4W1**: As per the suggestion of reviewer R4, we evaluate our proposed technique ARGS on larger dataset. In particular, We tried to run the PRBCD attack on the large OGBN-Products dataset, but we faced out-of-memory issues. We performed a more scalable attack GRBCD [1] on the OGBN-Products dataset. The OGBN-Products dataset contains 2,449,029 nodes and 61,859,140 edges, and performing attacks on this dataset is computationally intensive. We perform the GRBCD attack with a perturbation rate of 50%. We apply our proposed technique on this attacked graph to find ARGLTs and report the results in Table R1T4. We use GCN, GNNGuard, and Soft Median GDC as the baselines. From our preliminary experiments, we can observe that ARGS is able to find sparse ARGLTs for larger graphs, too.
>
> **R1T4** Graph Sparsity and Inference MACs of ARGLTs across different baselines.
> | Dataset | Perturbation Rate| Baseline | ARGLT | | |
> | --- | --- | --- | --- | --- | --- |
> ||||Graph Sparsity | Model Sparsity |Inference MACs |
> | OGBN-products  |50% (GRBCD Attack)  | GNNGuard | 30.17 % | 79.03 % |131,143 M |
> |       | | GCN | 36.97 % | 86.58 % |83,926 M |
> | || Soft Median GDC | 62.26 % | 98.56 % |9,005 M |
>  [1] Robustness of Graph Neural Networks at Scale, NeurIPS 2021
>
>
> **R4W2**: As per the suggestion of reviewer R4, we have revised the manuscript to fix any formatting issues. We have added the title for table 1 in the manuscript.
>
> **R4W3**: We recall that in the work "A Unified Lottery Ticket Hypothesis for Graph Neural Networks", the following steps are used to find the graph lottery tickets (GLTs):
>     1. Apply UGS to prune both the model and the graph during training and obtain the graph mask and model weight mask.
>     2. Rewound the GNN weights to their original initialization, apply the masks obtained in step 1 to the GNN weights and the graph, and train the pruned GNN model on the pruned graph, i.e., the GLT to obtain the performance of the GLT identified in this iteration.
>     3. Repeat steps 1 and 2 iteratively until the desired sparsity levels are reached.
>   We are also including the links to the codebase of UGS and specifying the lines of code where the above-mentioned steps are implemented.
>   Link:  https://github.com/VITA-Group/Unified-LTH-GNN/blob/main/NodeClassification/main_pruning_imp.py
>   Step 1: Line 168
>   Step 2:: Line 177
>
> Please note, that in our technique, we also follow the above-mentioned steps to identify the ARGLTs. The key difference lies in the fact that we apply ARGS instead of UGS for finding the graph and model weight masks in step 1 and the loss function used for training the GLT in step 2 in our case is different from the loss function used in the reference paper.

---

> > ### Author Response · Authors · 2023-11-20
> > **Response to reviewer eZKc (2/3)**
> >
> > **R4W4**: We thank the reviewer for pointing out these works on adversarial attacks and defenses for GNNs. We have included them in our related work section. We provide more details on each of these works:
> >
> >
> >   1. Adversarial Attacks on Node Embeddings via Graph Poisoning, ICML 2019: This work proposes poisoning attacks on a different task: unsupervised node representation learning (or node embeddings). The method exploits perturbation theory to maximize the loss obtained after training DeepWalk. In our work, we focus on semi-supervised learning. Therefore, this attack is orthogonal to our work.
> >
> >
> >   2. A Restricted Black-box Adversarial Framework Towards Attacking Graph Embedding Models, AAAI 2020: This work proposes a restricted black-box attack on graph embedding models. In the case of this attack, the attacker has access to only the training input and has limited knowledge of the model. In our work, we have considered a stronger attacker who has access to the training data, labels, and complete model architecture. Moreover, we consider an adaptive attack, which is also a stronger attack. For all these attacks, we observe that our technique is able to find highly sparse GLTs that are adversarially robust. Since these attacks are stronger, we believe that ARGS will also be effective for the restricted black-box attack.
> >
> >
> >   3. Not All Low-Pass Filters are Robust in Graph Convolutional Networks, NeurIPS 2021: This work proposes a spectral-based defense technique, GCN-LFR, that leverages the information that some low-frequency components in the graph spectrum are more robust to edge perturbations and regularizes the training process of a given GCN with robust information from an auxiliary regularization network to improve the adversarial performance of GCNs. We compare below this defense technique with ARGS and report the results in table R4T1. We also include other defense techniques for comparison. For the different defense baselines, the GLT which has similar accuracy as the baseline with maximum graph and model sparsity is identified as the ARGLT by ARGS. As evident from table R4T1, ARGS is able to identify ARGLTs that can achieve similar classification accuracy as the baseline with a very high level of sparsity and the total inference MAC of the ARGLTs is much less compared to the baselines.
> >
> >
> > **R4T1**: Performance comparison between ARGS and other defense techniques for the Citeseer dataset (GCN model)
> >
> >
> > |         | Citeseer (Mettack) Perturbation rate 10%    |     |
> > |-----------------|---------------------|-----------------------|
> > | Model                         | Accuracy (%)        | Inference MACs (M) |
> > | GCN-ProGNN                    | 69.94±0.45          | 3800.956     |
> > | GCN-ARGS                      | **70.2 ±0.59**      | **100.15**  |
> > | GCN-GNNGuard                  | 71.62±1.01          | 4101.87     |
> > | GCN-ARGS                      | **72.45±0.78**      | **210.59**  |
> > | GCN-GARNET                    | 71.97±1.20          | 3578.12     |
> > | GCN-ARGS                      | **72.45±0.78**      | **210.59**  |
> > | GCN-LFR                       | 69.1±0.14           | 4278.21     |
> > | GCN-ARGS                      | **70.2±0.59**      | **100.15**   |
> >
> >
> >
> >
> >   4. Robust Graph Representation Learning for Local Corruption Recovery, WWW 2023: This work proposes a learning scheme that automatically detects (locally) corrupted feature attributes and recovers robust embedding for different downstream tasks. Since we are considering structure perturbation attacks on graphs in our work, we regard this work on feature perturbation attacks as largely orthogonal to our work.

---

> ### Author Response · Authors · 2023-11-20
> **Response to reviewer eZKc (3/3)**
>
> **R4W5**: We have updated the adaptive attack section in our manuscript for better clarity. We are using the Meta-PGD attack [1] which is a gradient-based adaptive attack. Meta-PGD directly attacks ARGS by unrolling its training procedure. We have included the results of the adaptive attack on UGS in Table 2 in the manuscript and here in Table R4T2. We observe that ARGS is always able to find GLTs with much higher classification accuracy compared to UGS for the same sparsity levels. We report the accuracy of UGS and ARGS for 2 different sparsity levels and, in both cases, the accuracy of ARGS is better than UGS.
>
>
> **R4T2**: ARGS and UGS performance comparison for PGD attack and Adaptive Attack for Cora and Citeseer dataset (GCN model)
>
>
> | Dataset | Technique | Attack         | Graph Sparsity 22.64% / Model Sparsity 67.60% | Graph Sparsity 43.16% / Model Sparsity 91.70% | Graph Sparsity 22.64% / Model Sparsity 67.60% | Graph Sparsity 43.16% / Model Sparsity 91.70% |
> |---------|-----------|----------------|----------------------------------------------|----------------------------------------------|----------------------------------------------|----------------------------------------------|
> | Cora    | UGS       | PGD Attack     | 75.95±0.19                                   | 72.11±0.40                                   | 69.48±0.28                                   | 66.18±0.13                                   |
> |         |           | Adaptive Attack| 74.13±0.20                                   | 70.98±0.33                                   | 68.02±0.45                                   | 64.79±0.34                                   |
> | Cora    | ARGS      | PGD Attack     | **82.04±1.09**                               | **80.68±0.85**                               | **82.8±0.77**                                | **80.18±1.13**                               |
> |         |           | Adaptive Attack| **80.33±1.35**                               | **78.77±1.86**                               | **79.68±1.35**                               | **77.16±0.98**
> | Citeseer | UGS       | PGD Attack     | 74.63±0.30                                   | 72.29±0.36                                   | 70.70±0.26                                   | 69.34±0.49                                   |
> |          |           | Adaptive Attack| 69.8±0.31                                    | 66.97±0.25                                   | 65.48±0.31                                   | 63.98±0.18                                   |
> | Citeseer | ARGS      | PGD Attack     | **75.32±0.88**                               | **74.17±0.56**                               | **74.7±0.98**                                | **73.53±1.05**                               |
> |          |           | Adaptive Attack| **74.11±1.76**                               | **73.16±0.87**                               | **72.93±1.87**                               | **71.89±1.01**
>
>
> We apologize for the confusion created in Figure 6. The baseline here is GCN because it achieves better classification accuracy compared to GARNET for the OGBN-ArXiv dataset. As per the suggestion of reviewer R4, we now add Soft Median GDC as the baseline and compare the performance.
>
>
> In Table R4T3 we report the graph sparsity and model sparsity of the ARGLT obtained by ARGS for Cora and Citeseer datasets attacked by PGD (5% and 10% perturbation) for the Soft Median and STRG baselines.
>
>
> **R4T3**: PGD Attack
>
>
> | Dataset | Perturbation Rate| Baseline | ARGLT | |
> | --- | --- | --- | --- | --- |
> ||||Graph Sparsity | Model Sparsity |
> | Cora  |5% | STRG | 48.71% | 94.64% |
> |       |5% | Soft Median GDC | **51.28%** | **95.71%** |
> |       |10% | STRG |46.01% | 93.26% |
> |       |10% | Soft Median GDC | **60.42%** | **98.24%**|
> | Citeseer | 5% | STRG | 64.43% | 98.89% |
> | | 5% | Soft Median GDC | 64.43% | 98.89% |
> | | 10% | STRG | 43.19% | 91.61%|
> | | 10% | Soft Median GDC | **64.29%** | **98.89%** |
>
>
> As seen from Table R4T3, the ARGLTs obtained by ARGS for the Soft Median GDC baselines have equal or higher sparsity levels than the ARGLTs obtained by ARGS for the STRG baseline.

---

> > ### Author Response · Authors · 2023-11-22
> > **Kindly requesting response from reviewer eZKc**
> >
> > Dear Reviewer eZKc:
> >
> > We deeply appreciate your insightful comments and questions, which have significantly contributed to improve the quality of our manuscript. We look forward to hearing from you whether our answers, additional results, and modifications to the manuscript have satisfactorily addressed your concerns. We welcome any further feedback and we are happy to promptly make any further improvements to our submission.

---

### Official Review · Reviewer_69mY · 2023-10-31

**Soundness:** 3 good
**Presentation:** 2 fair
**Contribution:** 3 good
**Rating:** 6
**Confidence:** 3

**Summary:**

This paper presents the extension of the graph lottery ticket under the adversarial setting by leveraging the information from both generated pseudo labels and the smoothness regularization based on the observations from attacks on different types of graphs. Based on extensive experiments, the proposed algorithm discovers the existence of adversarially robust graph lottery tickets.

**Strengths:**

1. The proposed idea is described clearly.
2. The motivation from UGS and the analysis of the preliminary attack is clear.
3. The experiment shows that the proposed ARGS is better than the baseline UGS.

**Weaknesses:**

1. Some figures and tables are not referenced in the main content, e.g., Figure 2/4.
2. It seems unnecessary to have two hyper-parameters in Eq 8 after having the graph lottery ticket.
3. The proposed method seems incremental and uses a lot of hyper-parameters, which is difficult to tune in practice.

**Questions:**

1. What is the difference in the different rows of the untitled table of different GCN-ARGS? The performance is the same for Cora and different for Citeseer.
2. What is the performance of UGS under adaptive attacks? It is still necessary to know if the proposed method could match/be better than the baseline with the adaptive attacks.
3. There are multiple extensions of UGS. Is the proposed method better than them? For example, "Rethinking Graph Lottery Tickets: Graph Sparsity Matters. ICLR 2023"

---

> ### Author Response · Authors · 2023-11-20
> **Response to reviewer  69mY (1/2)**
>
> We reference the reviewer with the identifier 69mY as R3. We thank reviewer R3 for the comments and questions, which we address below. Comment n in the weaknesses is denoted by R3Wn while question n is denoted by R3Qn.
>
> **R3Q1**: We apologize for the confusion created by the table. In our updated manuscript, we have added the title. The table is now table 1 in the paper. In this table, we compare the performance of ARGS with other defense techniques, namely, ProGNN, GNNGuard, and GARNET. For these experiments, we have used GCN as the GNN model. When comparing with different defense techniques, the GLT which has the same classification accuracy as that of the defense technique with the maximum graph and model sparsity is identified as the adversarially robust graph lottery ticket (ARGLT) by ARGS. As mentioned in algorithm 1 in the manuscript, we iteratively prune the graph and the model until the target model and graph sparsity levels are reached. In our experiments, we prune the model parameters by 20% and graph edges by 5% at each iteration and set the maximum model sparsity to 98.9%. In cases where the classification accuracy of the defense technique is less than the classification accuracy of the GLT with maximum sparsity, we report this GLT as the ARGLT.
>
> In the case of the Cora dataset, the ARGLT identified by ARGS for PGD attack (20% perturbation rate) with maximum sparsity levels (model sparsity: 98.9%, graph sparsity: 64.1%) has a classification accuracy of 77.53%. The three different defense techniques, namely, ProGNN, GNNGuard, and GARNET have a classification accuracy of 63.43%, 73.19%, and 66.66%, respectively, which are all less than the classification accuracy of the most sparse ARGLT identified by ARGS. Therefore,  when comparing these different defense techniques, we report the performance of the ARGLT with maximum sparsity level.
>
> In the case of the Citeseer dataset attacked by MetaAttack with a 20% perturbation rate, the most sparse ARGLT has a classification accuracy of 70.2%. The defense technique ProGNN has a classification accuracy of 61.02%, which is less than the classification accuracy of the most sparse ARGLT. Hence, we compare it with this ARGLT. The defense techniques GNNGuard and GARNET have a classification accuracy of 71.62% and 72.97%, respectively. In these cases, the GLT with the same classification accuracy as the defense technique is reported in the table.
>
> **R3Q2**: We have included the performance of UGS under adaptive attack in Table 2 in the manuscript and here in Table R3T1. We observe that ARGS is always able to find GLTs with much higher classification accuracy compared to UGS for the same sparsity levels. We report the accuracy of UGS and ARGS for 2 different sparsity levels and, in both cases, the accuracy of ARGS is better than UGS.
>
> **R3T1**: ARGS and UGS performance comparison for PGD attack and Adaptive Attack for Cor and Citeseer dataset (GCN model)
> | Dataset | Technique | Attack         | Graph Sparsity 22.64% / Model Sparsity 67.60% | Graph Sparsity 43.16% / Model Sparsity 91.70% | Graph Sparsity 22.64% / Model Sparsity 67.60% | Graph Sparsity 43.16% / Model Sparsity 91.70% |
> |---------|-----------|----------------|----------------------------------------------|----------------------------------------------|----------------------------------------------|----------------------------------------------|
> | Cora    | UGS       | PGD Attack     | 75.95±0.19    | 72.11±0.40   | 69.48±0.28   | 66.18±0.13  |
> |         |           | Adaptive Attack| 74.13±0.20       | 70.98±0.3   | 68.02±0.45     | 64.79±0.34          |
> | Cora    | ARGS      | PGD Attack     | **82.04±1.09**                               | **80.68±0.85**                               | **82.8±0.77**                                | **80.18±1.13**                               |
> |         |           | Adaptive Attack| **80.33±1.35**                               | **78.77±1.86**                               | **79.68±1.35**                               | **77.16±0.98**
> | Citeseer | UGS       | PGD Attack     | 74.63±0.30                                   | 72.29±0.36                                   | 70.70±0.26                                   | 69.34±0.49                                   |
> |          |           | Adaptive Attack| 69.8±0.31                                    | 66.97±0.25                                   | 65.48±0.31                                   | 63.98±0.18                                   |
> | Citeseer | ARGS      | PGD Attack     | **75.32±0.88**                               | **74.17±0.56**                               | **74.7±0.98**                                | **73.53±1.05**                               |
> |          |           | Adaptive Attack| **74.11±1.76**                               | **73.16±0.87**                               | **72.93±1.87**                               | **71.89±1.01**

---

> > ### Author Response · Authors · 2023-11-20
> > **Response to reviewer 69mY (2/2)**
> >
> > **R3Q3**: There are extensions [1] of UGS but none of these works addresses the problem of finding Graph Lottery Tickets (GLT) when the input graphs may have been adversarially attacked. In particular, the effort in [1] tries to improve the performance of UGS for clean graphs and is orthogonal to our work. We observe that the proposed technique ARGS can also be applied to this improved version of UGS.
> >
> >
> > **R3W1**: We apologize for not including the caption for a table. We have now included it in the updated manuscript. Figures 2 and 4 have also been referenced in the original manuscript. We have highlighted these changes in the updated manuscript in blue.
> >
> >
> > **R3W2**: We note that the value of the two variables &eta; and &zeta; in equation 8 is always 1 and we do not change them. We are sorry for denoting them as hyperparameters. We have added this information to the manuscript.
> >
> >
> > **R3W3**: Our paper makes several contributions that we believe are significant. This is the first work that analyzes the adversarial robustness of GLTs and shows, with experimental results, that GLTs identified by UGS are indeed vulnerable to adversarial attacks. To this end, we propose a new technique ARGS that can find GLTs that are adversarially robust. Our proposed loss function is able to prune both malicious and irrelevant edges from the graph resulting in highly sparse GLTs that are resilient to adversarial attacks. Moreover, ARGS can be applied to both homophilic and heterophilic graphs making it a general technique for finding adversarially robust GLTs.
> >
> >
> > We observe that the proposed method has only three hyperparameters, namely, &beta; &lambda;1, and &lambda;2. We observe that the value of &alpha; and &gamma; in equation 7 is always 1 to enable uniform removal of edges surrounding the training as well as the test nodes. We have included these variables in the equation since they are set to 0 in the ablation study, to show the impact of different terms in the loss function on the performance of ARGS. Otherwise, we are sorry for incorrectly denoting these variables as hyperparameters. We have updated the manuscript accordingly.
> >
> >
> > [1] Rethinking Graph Lottery Tickets: Graph Sparsity Matters. ICLR 2023

---

> > > ### Author Response · Authors · 2023-11-22
> > > **Kindly requesting response from reviewer 69mY**
> > >
> > > Dear Reviewer 69mY:
> > >
> > > We deeply appreciate your insightful comments and questions, which have significantly contributed to improve the quality of our manuscript. We look forward to hearing from you whether our answers, additional results, and modifications to the manuscript have satisfactorily addressed your concerns. We welcome any further feedback and we are happy to promptly make any further improvements to our submission.

---

> > > > ### Comment · Reviewer_69mY · 2023-11-22
> > > >
> > > > Thank you for the effort in addressing my questions. Therefore, I have adjusted my score accordingly.

---

### Official Review · Reviewer_j8ef · 2023-10-31

**Soundness:** 3 good
**Presentation:** 3 good
**Contribution:** 3 good
**Rating:** 6
**Confidence:** 1

**Summary:**

This paper introduces an adversarially robust graph sparsification framework for creating highly sparse yet competitive Graph Lottery Tickets (GLTs). The authors investigate GLTs' resilience against structure perturbation attacks, demonstrating their superiority over dense graph neural networks in adversarial scenarios. They also propose a novel metric for assessing GLTs' robustness and validate the effectiveness of their framework on multiple benchmark datasets.

**Strengths:**

1. This paper tackles a crucial problem in graph neural networks by addressing their vulnerability to adversarial attacks. It introduces a novel framework for creating highly sparse yet competitive adversarially robust Graph Lottery Tickets (GLTs). This framework relies on a fresh loss function that captures graph homophily and information from both true train node labels and pseudo test node labels.

2. The paper extensively evaluates the proposed framework across various benchmark datasets, showcasing its effectiveness in producing resilient GLTs against different structure perturbation attacks. The authors also present a novel metric for assessing GLT robustness, considering model accuracy and sparsity.

3. The paper's clear and well-organized.

**Weaknesses:**

1. No theoretical analysis for the proposed method.

**Questions:**

1. How does the proposed framework handle dynamic graph structures?

---

> ### Author Response · Authors · 2023-11-20
> **Response to reviewer  j8ef**
>
> We reference the reviewer with the identifier eZKc as R2. We thank reviewer R2 for the comments, which we address below. We are happy to hear that you find our problem statement crucial and our proposed solution novel. Question n is denoted by R2Qn, and comment n in the weaknesses is denoted by R2Wn
>
> **R2W1**: In the current manuscript, our focus has been on a detailed empirical analysis of the proposed ARGS scheme both on mid and large-scale datasets. We refer to R1T4 for additional results on OGBN-products, which is a popular larger graph dataset. We believe such an empirical analysis of ARGS on various datasets (small, medium, and large-scale) is an important contribution to the machine learning scientific community and leave the theoretical analysis of the method as future research.
>
> **R2Q1**: In this work, we indeed consider graphs attacked by poisoning attacks. Poisoning attacks are train-time attacks where the graph is modified before the GNN model training, hence the graph does not have a dynamic structure. We leave the analysis of ARGS on dynamic graphs as future work.

---

> > ### Author Response · Authors · 2023-11-22
> > **Kindly requesting response from reviewer j8ef**
> >
> > Dear Reviewer j8ef:
> >
> > We deeply appreciate your insightful comments and questions. We look forward to hearing from you whether our answers, additional results, and modifications to the manuscript have satisfactorily addressed your concerns. We welcome any further feedback and we are happy to promptly make any further improvements to our submission.

---

### Official Review · Reviewer_3X3M · 2023-10-31

**Soundness:** 3 good
**Presentation:** 3 good
**Contribution:** 3 good
**Rating:** 6
**Confidence:** 3

**Summary:**

This paper studied the problem of finding Graph Lottery Tickets (GLT) when the input graphs may have been adversarially modified. The authors showed that graph pruning/sparsification can not only be used to reduce the model size and computational complexity, but also to stay robust to adversarially structural attacks. Extensive simulation results have shown the effectiveness of the proposed approach ARGS, and the implementation were provided as supplementary files.

**Strengths:**

The authors did a fairly comprehensive literature review of the existing methods on Graph Sparsification, Graph Adversarial Attacks, and their defenses. By analyzing the patterns of current attacking approaches (Figure 3), a new loss term $\mathcal{L}_{fs}$ is proposed to prune both malicious and irrelevant edges at the same time. Moreover, this loss term is universal for homophilic and heterophilic graphs, except that we need to consider the positional features instead of the node features for heterophilic graphs in this case. This is an interesting observation, and the importance of this term is verified by the ablation studies (Table 2). I am a little bit suspicious of the other term $\mathcal{L}_1$ though, where the authors use a new MLP trying to predict the test labels and use those with high confidence during training. Please see more details in the Questions part.

In summary, I think the observations and the method proposed in this paper are technically sound, and should be beneficial to the graph learning community.

**Weaknesses:**

Although this paper is good in general, there are indeed some weaknesses here.

Firstly, this paper only considered the structural attacks, instead of more practical attacking scenarios where node features and labels can also change. As a matter of fact, I think the design of $\mathcal{L}_{fs}$ and $\mathcal{L}_1$ heavily relies on the assumption that node features and labels would not change, which limits the application of the proposed approach. I am not sure what types of attacks are considered in Adaptive Attacks [1], but I do not see the intuition why ARGS would perform well when node features and graph topologies change at the same time.

Secondly, most datasets considered in the simulations are of small scale. Even OGB-ArXiv is also small, compared to other node classification benchmark datasets on OGB. I agree that it is totally okay to conduct simulations on standard datasets like the citation networks, and I guess the bottleneck would be running the attacks on the huge graphs, but I really do think it is much more interesting to perform this kind of task (finding robust GLT) on large-scale graphs where the graph + model sparsification is really needed.

[1] Felix Mujkanovic, Simon Geisler, Stephan Gunnemann, and Aleksandar Bojchevski. Are defenses for graph neural networks robust? Advances in Neural Information Processing Systems 35 (NeurIPS 2022), 2022.

**Questions:**

1. Although the focus of ARGS is to deal with graph perturbations, I am curious how well it can perform when there is no perturbation compared to UGS. Since ARGS can also be viewed as some kind of graph denoising, I would expect it to still outperform pure UGS in the normal training scheme.

2. How will Figure 3 look like if we also consider the positional features for homophilic graphs? If the pattern is still the same, why don't we just use the positional features for both homophilic and heterophilic graphs in $\mathcal{L}_{fs}$?

3. Is the term $\mathcal{L}_1$ still beneficial when we only have a very limited training set? And how shall we determine the confidence threshold for the test nodes?

---

> ### Author Response · Authors · 2023-11-20
> **Response to reviewer 3X3M (1/3)**
>
> We reference the reviewer with the identifier 3X3M as R1 and thank the reviewer for the helpful comments and insightful questions. We are happy to hear that you found our paper to be technically sound and to be beneficial for the graph learning community. We address the comments pointwise. Comment n in the weaknesses is denoted by R1Wn and question n is denoted by R1Qn.
>
> **R1Q1**: To answer this question, we ran experiments where we applied ARGS on clean graphs to find graph lottery tickets (GLTs). We report the results in table R1T1. ARGS is able to find highly sparse GLTs for clean graphs. Moreover, the lottery tickets found by ARGS achieve similar and often higher model and graph sparsity when compared to UGS for the same classification accuracy on Cora and Citeseer datasets across three different GNN models. We assume the accuracy of UGS at 0% graph and 0% model sparsity as the baseline accuracy.
>
> We recall that graph sparsity of 50% means that 50% of the graph edges are removed. Model sparsity of 50% means that 50% of the model parameters are removed.
>
> **R1T1**: Performance comparison between ARGS and UGS for clean graphs
> |  |  |       |  GCN  |            |    |  GIN      |            |   |  GAT      |            |
> |---------|------------------|---------------|------|---------|------|---------|---------------|-------|--------|---------------|
> | Dataset   | Technique | Accuracy |Graph Sparsity | Model Sparsity | Accuracy |Graph Sparsity | Model Sparsity  | Accuracy |Graph Sparsity | Model Sparsity |
> |Cora|UGS| 82.4±1.01% |18.5 % | 59.3 %| 80.05% | **5.0 %**  | **20.0 %** | 84.1%|46. 1% |93.4%|
> | |  ARGS | 82.4±0.89% | **25.9 %** | **67.59 %** | 80.05% | **5.0 %** | **20.0 %** |84.1%|**64.2 %**| **98.9%** |
> | Citeseer  | UGS |72.8±1.21% |48.8 % | 94.7% | 70.5% | 60.4 % | 98.2% | 74.8% | 51.3 % | 95.7% |
> |  |  ARGS | 72.8±1.17% | **64.3 %** |  **98.9%** | 70.5%| **64.3 %** | **98.9%**  |   74.8% | **60.4 %** |   **98.2%** |
>
> **R1Q2**:
> When we consider positional features of the nodes in homophilic graphs, we observe the overlap between the density distribution of positional feature differences of clean edges and that of adversarial edges is **higher** compared to that of the attribute features. However, the two density distributions of positional feature differences are still **separable**. To better understand how the two different feature types impact ARGS, we run a new set of experiments by considering the positional features of the nodes instead of the attribute features and by applying ARGS on the perturbed graphs to find GLTs. The results are reported in table R1T2. When positional features are considered, ARGS is still able to find highly sparse GLTs, but the sparsity levels of the GLTs in some cases are lower compared to the case when attribute features are considered.
>
> **R1T2** Evaluating the performance of ARGS when positional and attribute features are considered for homophilic graphs Cora and Citeseer on three different GNN models
> |  | |  | | GCN | | GIN |  | GAT | |
> |---------|--------|-------------------|------------------|---------------|---------------|---------------|---------------|---------------|---------------|
> | Dataset |  Attack  | Perturbation rate  | Feature type | Graph Sparsity | Model Sparsity | Graph Sparsity | Model Sparsity  | Graph Sparsity | Model Sparsity |
> | Cora     | PGD     | 5%  | Attribute    |   **49.2 %**  |  **94.8%**    |  **58.3 %**   |   **97.2%**    |   **64.2 %**  |   **98.9%**   |
> |   |    |   | Positional   |      46.0 %   |     93.3%     |  **58.3 %**   |   **97.2%**    |   **64.2 %**  |   **98.9%**   |
> |   |    | 10%  | Attribute    |   **48.3 %**  |  **94.2%**    |  **46.0 %**   |   **93.5%**    |   **64.3 %**  |   **98.9%**   |
> |    |   |   | Positional   |      37.2 %   |     87.1%     |    40.1 %     |   89.8%        |   **64.3 %**  |   **98.9%**   |
> | | | 15% | Attribute |   **49.1 %**  |  **94.8%**    |  **53.7 %**   |   **96.7%**    |   **64.2 %**  |   **98.9%**   |
> |  |  |  | Positional |      37.2 %   |    87.1%      |    18.5 %     |   59.2%        |   **64.3 %**  |   **98.9%**   |
> | Citeseer | Mettack | 5%  | Attribute |   **31.2 %**  |  **79.4 %**   |  **33.6 %**   |   **83.6%**    |   **64.2 %**  |   **98.9%**   |
> |  |  |  | Positional |  25.8 % | 74.2 % |  26.5 %       |     74.2%      |   **64.2 %**  |   **98.9%**   |
> |  |  | 10%| Attribute |   **33.6 %**  |  **83.6 %**   |  **64.2 %**   |   **98.8%**    |   **64.2 %**  |   **98.8%**   |
> |  |  |      | Positional   |   **33.6 %**  |  **83.6 %**   |  **64.2 %**   |   **98.8%**    |   **64.2 %**  |   **98.8%**   |
> |   |  | 15% | Attribute    |   **48.7 %**  |  **94.8%**    |  **64.2 %**   |   **98.8%**    |   **64.3 %**  |   **98.8%**   |
> |   |   |   | Positional   |     43.1 %    |    91.9%      |  **64.2 %**   |   **98.8%**    |   **64.3 %**  |   **98.8%**   |

---

> ### Author Response · Authors · 2023-11-20
> **Response to reviewer 3X3M (2/3)**
>
> **R1Q3**: In our experiments, the size of the training set is only 10% of the total number of nodes in the graph for the Cora, Citeseer, and PubMed datasets. This can be considered a limited training set size; for these datasets, we can observe from our ablation study that using the term **_L1_** is beneficial in improving the performance of ARGS. Therefore, we believe that, in limited training set size scenarios, the term **_L1_** is still beneficial. In our experiments, we set the value of the prediction confidence threshold to 0.9 to ensure that only the test nodes for which the model has high confidence are used to compute the CE loss term associated with the test nodes in equation 7 in the manuscript. We leave dynamic thresholding [1] as an interesting future research direction.
>
> [1] “Dash: Semi-supervised learning with dynamic thresholding.” ICML, 2021.
>
> **R1W1**: In our work, we have considered structure perturbation attacks on graphs which modify the graph topology by adding/removing edges from the graph. To evaluate how well ARGS performs when both the node features and the graph structure are attacked, we perform new experiments where we attack the graphs using a modified version of the Mettack (https://github.com/DSE-MSU/DeepRobust/blob/master/deeprobust/graph/global_attack/mettack.py). This attack perturbs both the node features and the graph structure. For a given perturbation budget, this attack at each iteration performs a structure perturbation and a feature perturbation, and between the two perturbations, the one that results in a higher attack loss is chosen. This iterative process is repeated until the perturbation budget is exhausted. We attack the Cora and Citeseer dataset with 5%, 10%, and 15% perturbation rates and use the STRG defense technique as the baseline. ARGS is able to find highly sparse GLTs that achieve similar classification accuracy as the baseline for different graph datasets perturbed with different perturbation rates using this attack. For example, for a 5% perturbation rate, ARGS finds GLT that have 53.75% graph sparsity and 96.55% model sparsity for the Cora dataset and 58.31% graph sparsity and 97.92% model sparsity for the Citeseer dataset. We add results for different perturbation rates in the appendix of the paper. Moreover, we compare the performance of ARGS with UGS in Table R1T3. For the same sparsity levels, GLTs identified by ARGS achieve much higher classification accuracy compared to GLTs identified by UGS. We observe that the attacked graph contains more edge perturbations than feature perturbations since modifying the graph structure results in higher attack loss than modifying the node features. This result shows that ARGS can find highly sparse GLTs for graphs attacked by both structure and node feature perturbations.
>
>
> **R1T3**: Performance comparison between ARGS and UGS for Cora and Citeseer datasets attacked by Mettack (modifying both structure and node features)
> |  | |  | GCN           | | | |
> |---------|--------------------------------------------|-------------------|----------------|--------------- |---------------|-------|
> | Dataset |  Attack                                    | Perturbation rate | Technique    | Accuracy (Graph Sparsity 22.64%, Model Sparsity 67.60%) | Accuracy (Graph Sparsity 43.16%, Model Sparsity 91.70%) |
> | Cora     | Mettack (Both structure and node feature) |     5%            |  UGS       |   75.50   |   72.23    |
> |          |                                           |                   |  ARGS      |   **83.35**   |    **81.04**   |
> |          |                                           |     10%           |  UGS       |   78.67   |    71.53   |
> |          |                                           |                   |  ARGS      |   **79.78**   |   **78.87**    |
> |          |                                           |    15%            |  UGS       |   63.18   |   60.41    |
> |          |                                           |                   |  ARGS      |   **80.03**   |    **77.01**   |
> | Citeseer | Mettack (Both structure and node feature) |     5%            |  UGS       |   71.86   |   71.09    |
> |          |                                           |                   |  ARGS      |   **76.07**   |    **75.71**   |
> |          |                                           |     10%           |  UGS       |   66.47   |   67.36    |
> |          |                                           |                   |  ARGS      |   **73.70**   |    **73.10**   |
> |          |                                           |    15%            |  UGS       |   67.24   |   65.58    |
> |          |                                           |                   |  ARGS      |   **71.56**   |   **69.31**    |

---

> > ### Author Response · Authors · 2023-11-20
> > **Response to reviewer 3X3M (3/3)**
> >
> > **R1W2**: We tried to run the PRBCD attack on the large OGBN-Products dataset, but we faced out-of-memory issues. We performed a more scalable attack GRBCD [1] on the OGBN-Products dataset. The OGBN-Products dataset contains 2,449,029 nodes and 61,859,140 edges and performing attacks on this dataset is computationally intensive. We perform the GRBCD attack with a perturbation rate of 50%. We apply our proposed technique on this attacked graph to find ARGLTs and report the results in table R1T4. We use GCN, GNNGuard, and Soft Median GDC as the baselines. From our preliminary experiments, we can observe that ARGS is able to find sparse ARGLTs for larger graphs too.
> >
> >
> > **R1T4** Graph Sparsity and Inference MACs of ARGLTs across different baselines.
> >
> > | Dataset | Perturbation Rate| Baseline | ARGLT | | |
> > | --- | --- | --- | --- | --- | --- |
> > ||||Graph Sparsity | Model Sparsity |Inference MACs |
> > | OGBN-products  |50% (GRBCD Attack)  | GNNGuard | 30.17 % | 79.03 % |131,143 M |
> > |       | | GCN | 36.97 % | 86.58 % |83,926 M |
> > | || Soft Median GDC | 62.26 % | 98.56 % |9,005 M |
> >  [1] Robustness of Graph Neural Networks at Scale, NeurIPS 2021

---

> > > ### Comment · Reviewer_3X3M · 2023-11-22
> > >
> > > Thank the authors for doing the new experiments and preparing the response. Regarding R1T1, it is a bit weird that the accuracies for UGS and ARGS are almost the same, while ARGS is sparser than UGS. What are the insights here? Where does this improvement mainly come from? This can help the readers better understand the paper.
> > >
> > > Also I believe the response for ICLR should be visible to everyone. No?

---

> > > > ### Author Response · Authors · 2023-11-22
> > > > **Follow up response to reviewer 3X3M**
> > > >
> > > > We thank the reviewer for the positive feedback on our responses and for the additional questions. Concerning Table **R1T1**, indeed, ARGS shows a performance improvement with respect to UGS on clean graphs as well. We believe this confirms the intuition previously advanced by the reviewer that ARGS can also be viewed as a kind of “graph denoising” method. In the case of ARGS, the loss formulation encourages removing irrelevant edges from a graph. Differently from UGS, ARGS considers not only the loss associated with the train nodes but also the one for the test nodes (L1). Therefore, ARGS tends to remove irrelevant edges surrounding both the train and the test nodes. This may help ARGS to achieve similar classification accuracy with higher sparsity levels. We agree with the reviewer that this insight would be useful for the readers and will add this to the revised version of our manuscript.
> > > >
> > > >
> > > > Concerning the visibility of our responses, we believe that the authors may choose the visibility of rebuttal responses to the reviewers during the review and rebuttal phases. Currently, our responses are visible to all the reviewers, ACs, SACs, and PCs. We would be happy to modify these settings if these are deemed as inappropriate at this time based on the ICLR policies.

---

> > > > > ### Author Response · Authors · 2023-11-22
> > > > > **Kindly requesting response from reviewer 3X3M**
> > > > >
> > > > > Dear reviewer 3X3M:
> > > > >
> > > > > We deeply appreciate your insightful comments and questions, which have significantly contributed to improve the quality of our manuscript. We look forward to hearing from you whether our answers, additional results, and modifications to the manuscript have satisfactorily addressed your concerns. We welcome any further feedback and we are happy to promptly make any further improvements to our submission.

---

### Author Response · Authors · 2023-11-20
**Official Comment by Authors**

We thank all of the reviewers for their thoughtful feedback and recognition of our paper’s contributions. We have addressed individual reviewers’ comments in their replies, as well as updated the manuscript PDF with the changes suggested. All changes in the PDF are marked in **blue** for ease of reference.

Summary of the different experiments we have conducted are:

* Evaluation of ARGS on OGBN-Products dataset ( having 2.5 million nodes and 61 million edges)
* Evaluation of ARGS on clean graph datasets
* Evaluation of ARGS on homophilic graphs with node positional features
* Evaluation of ARGS on  graphs whose structure and node features have been perturbed by the attack

We look forward to hearing your thoughts about our rebuttal, including whether it sufficiently addresses your concerns and questions.

---

### Meta-Review · Area_Chair_osg6 · 2023-12-18

**Metareview:**

This paper proposes a technique for reducing the vulnerability of graph lottery tickets to adversarial perturbations of the graph structure. The results appear to be reasonably effective on this problem. The reviewers raised several concerns, including the setting itself (are structure perturbations really the right threat model, and does focusing on that rely on other aspects not being attacked?), the complexity of the method itself (too many hyperparameters), and the sizes of the graphs being studied (are they big enough?). I share the first concern: is this really an important problem, and are these kinds of attacks the ones that matter in practice if adversarial attacks on graph lottery tickets are an important problem? I'm ambivalent about acceptance, and I lean toward rejection on the basis of the importance of the problem being studied. This seems like a reasonable contribution for this particular problem, but the problem itself is incredibly niche.

**Justification For Why Not Higher Score:**

I just can't, in good conscience, say that this is a problem we should care about. First, we have to care about GNNs. Then we have to care about lottery tickets within these GNNs. Then we have to care about adversarial attacks on those lottery tickets. Then we have to care specifically about adversarial attacks on the graph structure (rather than other, more relevant parts of the graph). I don't see the point.

**Justification For Why Not Lower Score:**

N/A

---

### Decision · Program_Chairs · 2024-01-16

Reject